# Parameterization-Based Dataset Distillation of 3D Point Clouds through Learnable Shape Morphing

**Dongwook Kim**[†]**, Jae-Young Yim**[†]**, and Jae-Young Sim**[⋆]
Graduate School of Artificial Intelligence
Ulsan National Institute of Science and Technology (UNIST)
{donguk071, yimjae0, jysim}@unist.ac.kr

## Abstract

Recent attempt in dataset distillation has been made to compress large-scale training datasets into compact synthetic versions, significantly reducing memory usage and training costs. While parameterization-based approaches have shown promising results on image datasets, their application to 3D point clouds remains largely unexplored due to the irregular and unordered nature of 3D data. In this paper, we first introduce a parameterization-based dataset distillation framework for 3D point clouds that enables the use of more diverse synthetic samples than conventional methods under the same memory budget. We construct an initial synthetic dataset containing multiple anchor samples with a coarser resolution than the original sample. We also generate new samples by morphing the shapes of the anchor samples with learnable weights to improve the diversity of the synthetic dataset. Moreover, we devise a uniformity-aware matching loss to ensure the structural consistency when comparing the original and synthetic datasets. Extensive experiments conducted on five standard benchmarks—ModelNet10, ModelNet40, ShapeNet, ScanObjectNN, and OmniObject3D—demonstrate that the proposed method effectively optimizes both the synthetic samples and the weights for shape morphing, outperforming existing dataset distillation methods.

## 1 Introduction

Significant advances in data-driven techniques for computer vision have been made possible by the availability of large-scale image datasets (Deng et al., 2009; Lin et al., 2014). However, training deep neural networks on large-scale datasets typically involves substantial computational costs and high memory consumption. To alleviate these issues, dataset distillation (Wang et al., 2018; Zhao & Bilen, 2023; Zhao et al., 2021a; Cazenavette et al., 2022; Zhang et al., 2024; Yim et al., 2025) has emerged as a promising approach for compressing large-scale datasets into compact yet representative synthetic datasets. Moreover, recent advances in the image domain have introduced a more storage-efficient paradigm known as distilled dataset parameterization (DDP).

DDP (Kim et al., 2022; Shin et al., 2023; Liu et al., 2022) represents the synthetic dataset in memory-efficient formats to synthesize a diverse and informative set of samples under the constrained storage budget. Specifically, some methods (Kim et al., 2022; Shin et al., 2023) attempt to reduce redundancy, allowing more synthetic samples to be represented within the same budget. This includes techniques such as removing spatial redundancy through downsampling (Kim et al., 2022) and suppressing less informative frequency components (Shin et al., 2023). In addition, other methods (Liu et al., 2022; Deng et al., 2022; Shin et al., 2025) adopt alternative representations, such as using generative models (Liu et al., 2022) to synthesize diverse training samples and neural fields (Shin et al., 2025) to represent datasets with a compact implicit function.

Large-scale 3D point cloud datasets have also enabled a wide range of applications in 3D vision (Zhao et al., 2021b; Yu et al., 2022; Park et al., 2022). However, only a lim-

---

[†]Equal contribution.     [⋆]Corresponding author.
Source code is available at https://github.com/yimjae0/3DDP

ited number of studies have developed dataset distillation methods tailored to 3D point clouds (Zhang et al., 2024; Yim et al., 2025). Furthermore, the parameterization techniques for 3D point clouds have not yet been explored, hindering the efficient use of storage space. In this paper, we first propose a parameterization-based dataset distillation framework for 3D point clouds that efficiently represent the synthetic dataset through learnable shape morphing.

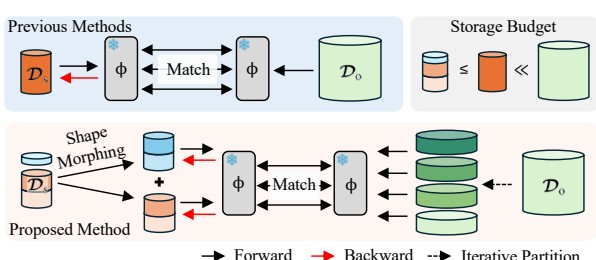

Figure 1 illustrates the conceptual difference between the proposed method and the previous methods. Whereas the previous methods directly optimize a synthetic point cloud dataset to match the original dataset, the proposed method parameterizes more diverse synthetic samples through learnable shape morphing. Specifically, we initialize the synthetic dataset by using multiple anchor samples at coarser resolutions, rather than representing a single full-resolution sample. We further extend the synthetic dataset including additional samples generated by blending the shapes of aligned anchor samples via learnable weights. This design enables the use of a larger number of samples within the same memory budget than the existing approaches. We jointly optimize the initial synthetic dataset and the set of learnable weights that minimize the uniformity-aware matching loss between the original and synthetic samples. We conduct extensive experiments to validate the effectiveness of our method, which consistently outperforms existing dataset distillation methods across all benchmarks.

Figure 1: The concept of the proposed distilled dataset parameterization approach compared to the existing dataset distillation approach.

The key contributions are summarized as follows.

- We are the first to propose a parameterization-based dataset distillation framework for 3D point clouds, which generates diverse synthetic samples under a constrained memory budget through learnable shape morphing.

- We jointly optimize the initial synthetic dataset as well as the learnable weights by minimizing a uniformity-aware matching loss between the partitioned original sample and the synthetic samples.

- We demonstrate that the proposed method achieves superior performance compared with existing dataset distillation techniques through extensive evaluations on standard 3D benchmarks, including ModelNet10 (Wu et al., 2015), ModelNet40 (Wu et al., 2015), ShapeNet (Chang et al., 2015), ScanObjectNN (Uy et al., 2019), and OmniObject3D (Wu et al., 2023).

## 2   RELATED WORK

**Dataset Distillation.**   Dataset distillation (Wang et al., 2018) was first formulated as a meta-learning problem, in which a small synthetic dataset is optimized to match the behavior of a model trained on the original dataset. Subsequent works (Zhao & Bilen, 2023; Zhao et al., 2021a; Cazenavette et al., 2022) have extended this idea in several directions. Gradient matching (Zhao et al., 2021a) aligns the training gradients of synthetic and original datasets, whereas trajectory matching (Cazenavette et al., 2022) extends this idea by matching the full training dynamics across multiple optimization steps. Distribution matching (Zhao & Bilen, 2023) matches feature distributions between the original and synthetic datasets, and achieves computational efficiency by avoiding the need to train a network during the distillation process. Recently, methods using generative models such as diffusion (Su et al., 2024) have also been explored for distilling informative samples.

**Point Cloud Dataset Distillation.**   Recently, dataset distillation has been extended to 3D point cloud data, which poses unique challenges due to their unordered and irregular nature. The earliest attempt, PCC (Zhang et al., 2024), applied a gradient-matching distillation framework, demonstrat-

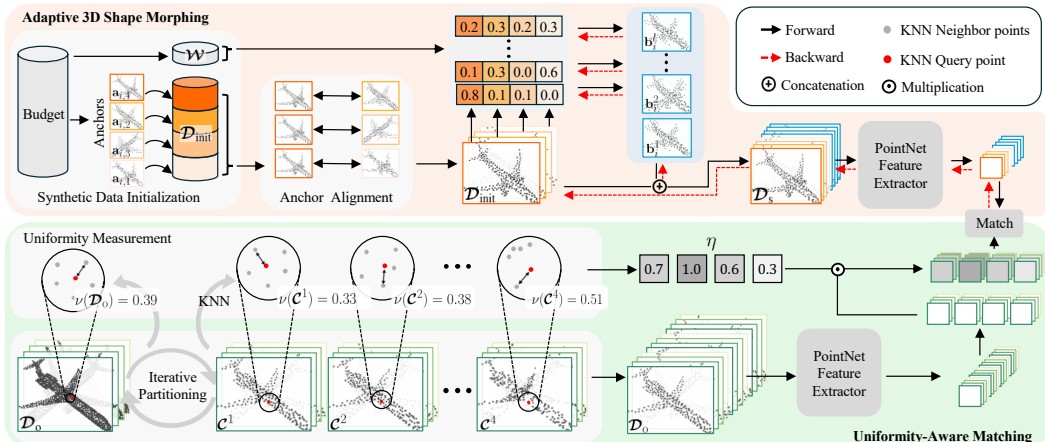

Figure 2: Overview of the proposed DDP framework for 3D point clouds. Adaptive 3D shape morphing increases the diversity of the synthetic dataset, while uniformity-aware matching ensures structural consistency between synthetic and original datasets.

ing the feasibility of dataset distillation in the 3D domain. A subsequent method, SADM (Yim et al., 2025), extends feature distribution matching to 3D point clouds by introducing a semantically aligned loss that explicitly accounts for their unordered structure. Additionally, it jointly optimizes the rotation angles, making the distillation process more robust to variations in orientation. DD3D (Bo & Wang, 2025) introduces a rotation-invariant dataset distillation framework for point clouds by combining a rotator with a point-wise generator, enabling resolution-flexible synthesis.

**Parameterization.** Dataset distillation parameterization aims to further reduce storage overhead by representing distilled data in specialized formats rather than as raw inputs. An early example, IDC (Kim et al., 2022), reduces storage cost by downsampling synthetic images to eliminate spatial redundancy, then upsampling them during training, allowing more samples to be stored under the same memory budget. FreD (Shin et al., 2023) performs dataset distillation in the frequency domain, discarding less important frequency components to reduce redundancy. This frequency-level compression allows more synthetic samples to be used under the same memory budget while preserving global structure. Different strategies, such as HaBa (Liu et al., 2022) and DDiF (Shin et al., 2025), adopt alternative parameterization strategies for efficient storage of synthetic datasets. HaBa employs a generative parameterization that distills data in a discrete latent space instead of the raw pixel space. In contrast, DDiF represents each synthetic instance as a neural field, which is a continuous function that maps coordinates to data values. Other methods (Deng et al., 2022; Wei et al., 2023) aim to represent synthetic datasets more efficiently by capturing shared patterns across data, rather than treating each sample independently.

## 3 METHODOLOGY

We propose a parameterization-efficient dataset distillation framework for 3D point clouds. As shown in Figure 2, it comprises two core components: adaptive shape morphing and uniformity-aware matching.

### 3.1 PROBLEM FORMULATION

**Dataset Distillation.** Let $\mathcal{D}_o = \{\mathbf{x}_i\}_{i=1}^{O}$ denote the original dataset and $\mathcal{D}_s = \{\mathbf{s}_i\}_{i=1}^{S}$ denote the synthetic dataset, where $S \ll O$. The goal of dataset distillation (DD) is to generate an optimal synthetic dataset $\mathcal{D}_s^*$ such that a model trained on $\mathcal{D}_s^*$ exhibits similar behavior to that trained on $\mathcal{D}_o$. In practice, the optimization problem can be formulated as

$$\mathcal{D}_s^* = \underset{\mathcal{D}_s}{\arg\min}\, \mathcal{L}(\mathcal{D}_o, \mathcal{D}_s), \tag{1}$$

where $\mathcal{L}$ is a matching loss that measures the discrepancy between the original and synthetic datasets. Depending on how it is defined, the DD employs different approaches, such as feature distribution matching, gradient matching, or training trajectory matching.

**Distilled Dataset Parameterization.** DDP represents the synthetic dataset in a more compact form using a set of latent codes $\mathcal{Z} = \{\mathbf{z}_i\}_{i=1}^{Z}$ and the parameters $\theta$ of a decoder $g_\theta$, such that the synthetic dataset is defined as

$$\mathcal{D}_{\mathrm{s}} = \{g_\theta(\mathbf{z}_i)\}_{i=1}^{Z}. \tag{2}$$

By storing the latent codes and the parameters of decoder rather than storing the synthetic samples directly, DDP enables efficient use of memory allowing a larger number of synthetic samples to be utilized under the same storage budget. Therefore, the objective of DDP is to jointly optimize the latent codes $\mathcal{Z}^*$ and the decoder parameters $\theta^*$ such that the discrepancy between the original and synthetic datasets is minimized.

$$\{\mathcal{Z}^*, \theta^*\} = \underset{\{\mathcal{Z}, \theta\}}{\operatorname{argmin}} \mathcal{L}(\mathcal{D}_{\mathrm{o}}, \mathcal{D}_{\mathrm{s}}). \tag{3}$$

## 3.2 Synthetic Dataset Parameterization through Learnable Shape Morphing

To increase the diversity of the synthetic dataset within the constrained memory budget, we propose a distilled dataset parameterization method of 3D point clouds that utilizes additional synthetic samples generated by learnable shape morphing. As illustrated by the adaptive 3D shape morphing module in Figure 2, we randomly sample 3D point cloud objects from the original dataset to initialize the synthetic dataset. Instead of selecting an original (full-resolution) sample with $N_1$ points, we take $M$ distinct coarser samples, called anchors, each containing $N_2$ points. The set of these $M$ anchors is referred to as a group. Then we construct an initial synthetic dataset as

$$\mathcal{D}_{\mathrm{init}} = \left\{\{\mathbf{a}_{i,m}\}_{m=1}^{M}\right\}_{i=1}^{S}, \tag{4}$$

where $\mathbf{a}_{i,m} \in \mathbb{R}^{N_2 \times 3}$ denotes the $m$-th anchor sample in the $i$-th group. To ensure that the total memory budget of $M$ anchors is smaller than the full-resolution one, we set the constraint such that $MN_2 \leq N_1$.

Inspired by 3D shape morphing, we generate additional point cloud samples by blending the shapes of the selected anchors to further enhance the diversity of the synthetic dataset. Specifically, we first establish point-wise correspondences across the anchor samples. For each $i$-th group, we align the anchor samples to the first anchor $\mathbf{a}_{i,1}$. We construct the pairwise Euclidean distance matrix between $\mathbf{a}_{i,1}$ and each of the remaining $M-1$ anchor samples, and solve a linear assignment problem to obtain one-to-one correspondence. Then the points in each sample are reordered according to the resulting correspondences. We interpolate $L$ additional samples from the $M$ re-ordered anchor samples by computing convex combination with learnable weights that adaptively control the contribution of the anchors. Specifically, the $l$-th new sample $\mathbf{b}_i^l \in \mathbb{R}^{N_2 \times 3}$ in the $i$-th group is obtained by blending the shapes of the re-ordered anchors $\tilde{\mathbf{a}}_{i,m}$'s as

$$\mathbf{b}_i^l = \sum_{m=1}^{M} w_{i,m}^l \cdot \tilde{\mathbf{a}}_{i,m}, \tag{5}$$

using a learnable weight vector $\mathbf{w}_i^l = \left[w_{i,1}^l, \ldots, w_{i,M}^l\right]$ such that $\sum_{m=1}^{M} w_{i,m}^l = 1$ and $w_{i,m}^l \geq 0$. Although interpolation is performed over the aligned point cloud samples, perfect point-wise correspondences are not guaranteed due to dataset-specific variations, such as random rotations around the up-axis. Thus we optimize each learnable weight vector $\mathbf{w}_i^l$ in an adaptive manner to mitigate such potential mismatches. Note that this strategy introduces no additional memory cost as it reuses the existing anchors.

Finally, we merge the initial synthetic dataset $\mathcal{D}_{\mathrm{init}}$ with the set of the combined samples to construct a complete synthetic dataset $\mathcal{D}_{\mathrm{s}}$.

$$\mathcal{D}_{\mathrm{s}} = \left\{\{\tilde{\mathbf{a}}_{i,m}\}_{m=1}^{M} \cup \{\mathbf{b}_i^l\}_{l=1}^{L}\right\}_{i=1}^{S}. \tag{6}$$

Note that the conventional DD setting uses only a single full-resolution synthetic sample of $\mathbf{s}_i$, however the proposed DDP method facilitates the use of $M$ times more diverse shapes of anchor samples as well as $L$ additional combined samples through the learnable convex combination, expanding the diversity of synthetic dataset.

### 3.3 DATASET DISTILLATION WITH UNIFORMITY-AWARE MATCHING LOSS

We perform the dataset distillation based on the feature distribution matching by adopting the SADM loss (Yim et al., 2025), that matches semantically aligned feature distributions between the original and synthetic datasets, defined as

$$\mathcal{L}_{\text{SADM}}(\mathcal{D}_{\text{o}}, \mathcal{D}_{\text{s}}) = \tilde{\mathcal{K}}_{\mathcal{D}_{\text{o}}, \mathcal{D}_{\text{o}}} + \tilde{\mathcal{K}}_{\mathcal{D}_{\text{s}}, \mathcal{D}_{\text{s}}} - 2\tilde{\mathcal{K}}_{\mathcal{D}_{\text{o}}, \mathcal{D}_{\text{s}}}, \tag{7}$$

where $\tilde{\mathcal{K}}_{\mathcal{D}_{\text{o}}, \mathcal{D}_{\text{s}}}$ denotes the kernel function computed over the sorted feature representations. However, SADM assumes that samples from $\mathcal{D}_{\text{o}}$ and $\mathcal{D}_{\text{s}}$ share the same resolution, which is not the case in our setting, where $\mathcal{D}_{\text{s}}$ includes more coarsely sampled points than the full-resolution samples in $\mathcal{D}_{\text{o}}$. Therefore, we partition each sample $\mathbf{x}_i$ in $\mathcal{D}_{\text{o}}$ into $M$ non-overlapping low-resolution samples by iteratively applying the farthest point sampling (FPS), where each low-resolution sample contains $N_2$ points. Then we gather the $m$-th partitioned samples over all the original samples to construct the corresponding subset $\mathcal{C}^m$, which are compared to the synthetic dataset $\mathcal{D}_{\text{s}}$.

Note that the resulting subsets of $\mathcal{C}^1, \mathcal{C}^2, \ldots, \mathcal{C}^M$ may exhibit spatial non-uniformity of point distributions, that may degrade the reliability of distribution matching. We adaptively control the contribution of subsets to the loss computation according to their uniformity. Specifically, as shown in the uniformity-aware matching module of Figure 2, we estimate the uniformity score $\nu(\mathcal{D})$ of the dataset $\mathcal{D}$ by using the average coefficient of variation (CV) of the local distances computed across the $k$ nearest neighbors.

$$\nu(\mathcal{D}) = \frac{1}{N(\mathcal{D}) \cdot O} \sum_{i=1}^{O} \sum_{j=1}^{N(\mathcal{D})} \frac{\sigma_j^i}{\mu_j^i + \epsilon}, \tag{8}$$

where $\mu_j^i$ and $\sigma_j^i$ denote the mean and standard deviation of the distances from the $j$-th point in the $i$-th sample to its $k$ nearest neighbors, respectively, and $\epsilon$ is a small constant for numerical stability. $N(\mathcal{D})$ is the number of points in each sample, which is identical across all samples in $\mathcal{D}$. Then the penalty of $\mathcal{C}^m$ is estimated by

$$\eta^m = \exp\left(-\lambda \left(\nu(\mathcal{D}_{\text{o}}) - \nu(\mathcal{C}^m)\right)^2\right), \tag{9}$$

where $\lambda$ is a scaling parameter.

Finally, the uniformity-aware distribution matching loss for DD is designed as follows:

$$\mathcal{L}_{\text{Distill}}(\mathcal{D}_{\text{o}}, \mathcal{D}_{\text{s}}) = \sum_{m=1}^{M} \eta^m \cdot \mathcal{L}_{\text{SADM}}(\mathcal{C}^m, \mathcal{D}_{\text{s}}). \tag{10}$$

Then the overall optimization objective is to jointly optimize the initial synthetic dataset $\mathcal{D}_{\text{init}}^*$ and the set of learnable weights $\mathcal{W}^*$ to minimize the distillation loss $\mathcal{L}_{\text{Distill}}(\mathcal{D}_{\text{o}}, \mathcal{D}_{\text{s}})$:

$$\{\mathcal{D}_{\text{init}}^*, \mathcal{W}^*\} = \underset{\{\mathcal{D}_{\text{init}}, \mathcal{W}\}}{\text{argmin}} \, \mathcal{L}_{\text{Distill}}(\mathcal{D}_{\text{o}}, \mathcal{D}_{\text{s}}), \tag{11}$$

where $\mathcal{W} = \left\{\{\mathbf{w}_i^l\}_{l=1}^{L}\right\}_{i=1}^{S}$, $\mathcal{D}_{\text{init}}$ is defined in (4), and $\mathcal{D}_{\text{s}}$ is defined in (6).

### 3.4 STORAGE BUDGET ANALYSIS

In the conventional DD setting, each synthetic sample is stored at full-resolution with $N_1$ points, where the coordinates of each point are represented by three 32-bit floating-point numbers, requiring $96N_1$ bits per sample. Assuming $K$ point clouds per class (PPC) for $C$ classes, the total memory budget is constrained to $96N_1KC$ bits. On the other hand, the proposed method maintains this budget by representing each synthetic sample using $M$ coarser anchor samples, each containing $N_2$ points, resulting in the storage cost of $96MN_2KC$ bits. Also, the learnable shape morphing further enhances the diversity incurring only a small overhead of $32L(M-1)KC$ bits to store the learnable weights. Hence the total memory constraint for the proposed method is

$$96MN_2KC + 32L(M-1)KC \leq 96N_1KC. \tag{12}$$

Note that the weight storage term is proportional to $M-1$ rather than $M$, because one weight is determined by the condition $\sum_{m=1}^{M} w_{i,m}^l = 1$.

Table 1: Classification performance of the proposed method compared with coreset selection and dataset distillation methods. All methods were evaluated using PointNet under the same memory budget. 'Whole' indicates the result obtained by training on the entire original dataset. The best performance in each row is highlighted in bold. **OOM** denotes out of memory during distillation.

| Dataset | PPC | Coreset Selection | | | Dataset Distillation | | | | | | Whole |
| | | Random | Herding | K-Center | DM | DC | MTT | PCC | SADM | Ours | |
|---|---|---|---|---|---|---|---|---|---|---|---|
| ModelNet10 | 1 | 28.1±4.2 | 34.0±6.5 | 34.0±6.5 | 25.8±6.9 | 32.8±8.5 | 27.8±5.8 | 33.0±8.0 | 35.9±8.2 | **87.7±0.7** | |
| | 3 | 74.5±1.8 | 76.9±1.2 | 75.9±1.8 | 77.4±1.2 | 74.5±2.6 | 73.6±1.7 | 70.7±1.6 | 83.5±0.7 | **89.8±0.5** | 92.18 |
| | 10 | 84.7±0.7 | 86.1±0.7 | 82.2±1.5 | 85.0±0.7 | 84.6±0.6 | 85.3±1.2 | 86.3±1.1 | 87.4±1.1 | **92.2±0.5** | |
| ModelNet40 | 1 | 34.0±2.1 | 54.1±2.1 | 54.1±2.1 | 31.1±4.7 | 50.3±2.0 | 33.4±2.1 | 55.3±1.4 | 54.8±1.3 | **73.2±1.1** | |
| | 3 | 59.9±1.6 | 69.1±1.0 | 62.1±2.7 | 61.5±2.1 | 66.0±1.1 | 59.5±0.6 | 66.2±1.6 | 71.3±0.7 | **80.3±0.5** | 88.78 |
| | 10 | 73.3±0.9 | 77.6±0.6 | 64.3±1.3 | 74.9±0.8 | 74.3±0.9 | 73.4±0.5 | 77.9±0.9 | 79.6±0.6 | **82.5±0.6** | |
| ShapeNet | 1 | 33.5±2.5 | 49.1±2.4 | 49.1±2.4 | 26.3±3.6 | 48.7±1.6 | 32.4±2.6 | 50.9±3.5 | 51.1±2.3 | **60.5±1.1** | |
| | 3 | 53.4±1.4 | 58.8±1.0 | 50.6±1.6 | 52.5±1.6 | 56.6±1.1 | 53.5±2.0 | 58.9±1.7 | 62.2±1.6 | **65.9±0.6** | 82.49 |
| | 10 | 62.4±0.9 | 66.3±0.4 | 46.9±0.7 | 63.1±0.8 | 63.7±0.8 | 62.3±1.1 | 65.4±0.8 | 68.0±0.5 | **68.9±0.6** | |
| ScanObjectNN | 1 | 13.5±1.8 | 15.1±1.7 | 15.1±1.7 | 13.7±1.8 | 15.2±2.0 | 14.3±2.5 | 16.0±2.4 | 17.6±1.5 | **32.6±1.6** | |
| | 3 | 19.7±0.7 | 26.9±1.4 | 18.8±1.1 | 26.4±2.4 | 24.6±2.2 | 20.1±1.3 | 25.5±2.2 | 32.6±1.6 | **41.3±1.1** | 63.43 |
| | 10 | 34.1±1.6 | 38.3±1.6 | 23.5±0.9 | 37.4±1.2 | 38.5±1.6 | 37.1±2.0 | 34.6±1.4 | 43.7±2.0 | **49.8±0.7** | |
| OmniObject3D | 1 | 24.0±0.6 | 30.5±1.0 | 30.5±1.0 | 15.1±0.9 | 31.1±0.5 | 25.8±1.3 | 35.8±0.6 | 33.2±0.4 | **41.9±1.3** | |
| | 3 | 40.9±1.2 | 42.3±1.4 | 35.9±1.5 | 40.3±1.3 | 44.3±1.0 | 44.8±1.1 | 46.2±0.7 | 51.6±0.4 | **58.4±1.5** | 74.98 |
| | 10 | 59.5±2.1 | 59.3±1.7 | 58.3±1.6 | 60.0±1.6 | 56.5±0.8 | **OOM** | 57.0±0.6 | 65.2±1.7 | **68.4±1.2** | |

## 4 EXPERIMENTAL RESULTS

### 4.1 EXPERIMENTAL SETUPS

**Datasets.** We evaluate the performance of the proposed method on five standard point cloud classification datasets: ModelNet10 (Wu et al., 2015), ModelNet40 (Wu et al., 2015), ShapeNet (Chang et al., 2015), ScanObjectNN (Uy et al., 2019), and OmniObject3D (Wu et al., 2023). ModelNet10 and ModelNet40 consist of 10 and 40 categories of clean 3D CAD models, respectively. ShapeNet includes 55 categories of large-scale 3D CAD models with finer-grained class distinctions. ScanObjectNN comprises 15 categories of real-world objects captured from RGB-D scans, and we use the PB_T50_RS variant, which is the most challenging setting in this dataset. To further validate the scalability of our method, we conducted additional experiments on OmniObject3D (Wu et al., 2023), a dataset that contains a significantly larger number of object categories compared to conventional benchmarks. Since OmniObject3D does not provide an official train–test split, we randomly sampled 80% of the data for distillation and used the remaining 20% for testing, while ensuring that each class included at least four test samples. This resulted in a total of 156 categories used in our evaluation. We additionally evaluate part segmentation performance using ShapeNetPart (Yi et al., 2016).

**Implementation Details.** Both $\mathcal{D}_{init}$ and $\mathcal{W}$ were updated via stochastic gradient descent with a learning rate of 10 for 2,000 iterations. Training was performed for 500 epochs with a batch size of 8, using a step decay schedule with a step size of 250 and a decay rate of 0.1. All reported results were averaged over 10 independent runs by using a single NVIDIA RTX 3090 GPU. To ensure a strictly fair comparison, all baselines were fully re-implemented and evaluated under an identical and augmentation-free setting.

### 4.2 PERFORMANCE COMPARISON

We compared the performance of the proposed method with representative dataset distillation methods, including DM (Zhao & Bilen, 2023), DC (Zhao et al., 2021a), and MTT (Cazenavette et al., 2022), which were originally developed for image domains and adapted to 3D point clouds. We also compared SADM (Yim et al., 2025), and PCC (Zhang et al., 2024), recent methods tailored to 3D point cloud dataset distillation. In addition, we compared coreset selection methods commonly used to reduce dataset size, including random selection (Rebuffi et al., 2017), Herding (Castro et al., 2018), and K-Center (Sener & Savarese, 2018).

Table 2: Comparison of cross-architecture generalization performance at PPC = 1, evaluated on PointNet++(Qi et al., 2017b) (PN++), PointConv(Wu et al., 2019) (PC), Point Transformer (Zhao et al., 2021b) (PT), and PointMamba (Liang et al., 2024) (PM). The best performance in each row is highlighted in bold.

| Dataset | Method | Random | DM | DC | MTT | PCC | SADM | Ours |
|---|---|---|---|---|---|---|---|---|
| ModelNet10 | PN++ | 22.4±6.9 | 12.1±2.9 | 15.3±6.5 | 20.4±6.9 | 20.7±6.1 | 25.9±7.1 | **55.4±8.6** |
| | PC | 17.7±10.1 | 10.8±3.6 | 14.2±4.7 | 21.3±7.6 | 16.0±8.7 | 20.5±13.1 | **51.6±9.8** |
| | PT | 44.1±6.3 | 22.4±9.1 | 26.7±6.9 | 39.3±7.4 | 45.9±7.5 | 49.0±7.7 | **57.0±10.9** |
| | PM | 29.2±8.1 | 14.0±2.1 | 20.7±2.4 | 31.6±4.9 | 28.9±7.3 | 28.4±3.7 | **69.4±1.6** |
| ModelNet40 | PN++ | 36.8±2.2 | 1.5±1.3 | 8.6±3.0 | 37.2±1.8 | 13.5±3.3 | 40.0±2.8 | **47.7±5.0** |
| | PC | 23.1±3.8 | 3.9±1.9 | 11.1±3.4 | 24.2±4.6 | 14.7±3.8 | 29.1±3.0 | **33.2±5.9** |
| | PT | 28.9±1.2 | 6.2±4.4 | 14.8±3.1 | 29.0±1.2 | 40.2±2.2 | **44.5±1.3** | 39.0±6.6 |
| | PM | 34.1±1.3 | 12.3±2.2 | 24.3±1.6 | 33.4±1.4 | 38.3±2.4 | 35.9±2.0 | **59.6±0.9** |
| ShapeNet | PN++ | 25.3±2.5 | 2.0±1.4 | 7.2±2.0 | 24.6±2.5 | 21.8±3.2 | 35.1±1.3 | **44.7±2.0** |
| | PC | 19.0±3.1 | 3.8±1.0 | 10.3±3.4 | 19.4±3.9 | 16.5±4.4 | 20.3±5.0 | **24.3±6.0** |
| | PT | 26.3±1.3 | 7.4±2.9 | 19.1±4.9 | 26.3±1.7 | 38.3±1.6 | 36.3±2.6 | **40.2±7.6** |
| | PM | 17.4±1.3 | 5.6±1.3 | 13.1±1.6 | 17.3±1.4 | 30.7±1.1 | 25.6±3.1 | **49.0±0.7** |
| ScanObjectNN | PN++ | 18.0±1.4 | 14.5±4.7 | 15.8±3.7 | **18.8±2.6** | 13.0±3.6 | 9.3±2.1 | 14.3±2.5 |
| | PC | 12.4±2.2 | 10.0±2.1 | 9.9±1.8 | 13.1±1.8 | 12.0±2.5 | 10.9±3.4 | **14.6±2.1** |
| | PT | 12.5±0.9 | 10.2±2.2 | 12.3±1.8 | 12.2±1.6 | 16.2±1.5 | 15.8±2.0 | **17.6±1.8** |
| | PM | 18.9±1.6 | 14.8±1.7 | 17.7±1.7 | 18.9±1.4 | 13.6±2.4 | 13.2±0.9 | **19.7±1.0** |

Table 3: Comparison of part segmentation performance on the ShapeNet dataset with PPC set to 1. K-Center results are omitted from the comparison because they are identical to Herding when the PPC is 1.

| Class | Air. | Bag | Cap | Car | Chair | Ear. | Guitar | Knife | Lamp | Laptop | Motor. | Mug | Pistol | Rocket | Skate. | Table | Avg. |
|---|---|---|---|---|---|---|---|---|---|---|---|---|---|---|---|---|---|
| Whole | 82.0 | 65.5 | 65.3 | 75.0 | 88.6 | 68.2 | 90.2 | 83.0 | 77.7 | 94.9 | 63.0 | 92.8 | 79.0 | 53.9 | 70.5 | 81.3 | 76.9 |
| Random | 28.1 | 22.8 | 53.0 | 21.6 | 38.8 | 24.0 | 45.0 | 23.1 | 25.1 | 57.1 | 20.3 | 46.0 | 36.4 | 29.7 | 13.7 | 24.1 | 31.8 |
| Herding | 31.7 | 36.1 | 47.1 | 22.1 | 46.4 | 34.9 | 50.4 | 54.2 | 20.9 | 65.2 | 16.3 | 54.9 | 35.0 | 30.8 | 31.5 | 42.3 | 38.7 |
| SADM | 29.9 | 30.6 | 52.3 | 21.6 | 49.4 | 23.1 | 51.3 | 66.0 | 29.3 | 68.8 | 15.0 | 51.7 | 39.8 | 35.4 | 38.5 | 46.5 | 40.6 |
| Ours | **51.5** | **51.8** | **59.0** | **36.5** | **70.8** | **42.6** | **80.1** | **76.6** | **31.8** | **81.0** | **26.4** | **84.3** | **61.8** | **40.5** | **49.2** | **59.1** | **56.4** |

**Evaluation on PointNet.** Table 1 compares the classification performance of different methods using PointNet (Qi et al., 2017a) under the same memory budget. Specifically, we set $N_2 = 252$, $M = 4$, and $L = 16$ for ModelNet10 (Wu et al., 2015), and $N_2 = 255$, $M = 4$, and $L = 4$ for the other datasets, respectively, to satisfy the inequality in (12) with $N_1 = 1024$ for the original datasets. The proposed method consistently outperforms all compared methods across all benchmark datasets and PPC settings. In particular, the most substantial performance gain is observed when PPC is set to 1. For instance, on ModelNet10 at PPC = 1, our method achieves an accuracy of 87.7%, which is a remarkable improvement over 35.9%, the state-of-the-art (SOTA) performance of SADM. Similarly, on ModelNet40, our method reaches 73.2%, outperforming all the baselines by a large margin. Moreover, our method improves the state-of-the-art performance from 17.6% to 32.6% demonstrating its reliability on the challenging real-world dataset ScanObjectNN at PPC = 1. On OmniObject3D, which includes a substantially larger and more fine-grained set of categories, our method achieves 41.9%, indicating that the proposed framework generalizes reliably even as the number of categories increases significantly. These results demonstrate that the proposed parameterization technique provides more promising approach for dataset distillation of 3D point clouds under constrained memory budgets than the existing methods.

**Cross-Architecture Generalization.** To evaluate the cross architecture generalization performance, we compared the performance of the proposed method with the existing methods using four different architectures including PointNet++ (Qi et al., 2017b), PointConv (Wu et al., 2019), Point Transformer (Zhao et al., 2021b), and PointMamba (Liang et al., 2024), after distillation is performed using PointNet (Qi et al., 2017a). As summarized in Table 2, the proposed method consistently achieves the best performance across the most datasets and architectures, demonstrating strong generalization ability. On ModelNet10, the proposed method achieves 55.4% with PointNet++, significantly outperforming 25.9%, the performance of SADM. Likewise, on PointConv, Point Transformer and PointMamba, the proposed method provides 51.6%, 57.0%, and 69.4%,

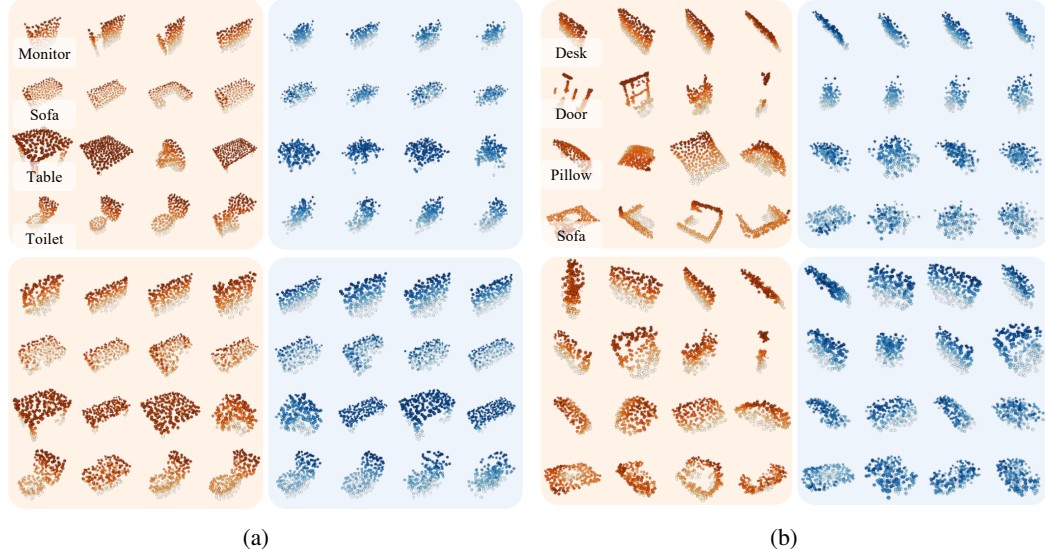

(a)                                                    (b)

Figure 3: Visualization of the resulting synthetic datasets: the first four rows show initial samples before optimization, consisting of anchor samples (orange) and combined samples (blue), while the subsequent four rows illustrate samples obtained after applying the proposed method. (a) Model-Net10 and (b) ScanObjectNN.

respectively, maintaining substantial performance gaps over all the baselines. Similar trends of improvement are observed on ModelNet40 and ShapeNet. While our method generally improves the performance across all architectures, the accuracy on ScanObjectNN with PointNet++ is slightly lower. This is because our method designs the dataset distillation loss based on SADM loss (Yim et al., 2025), which shows relatively low performance on ScanObjectNN with PointNet++. These results demonstrate that the synthetic samples, generated by combining diverse low-resolution anchors in the proposed method, are not overfitted to specific architectures and instead capture useful geometric characteristics that generalize well across different backbone networks.

**Part Segmentation Evaluation.**    To validate the generalization of the proposed method beyond classification, we additionally performed a part segmentation experiment on the ShapeNetPart (Yi et al., 2016) using a PointNet segmentation model. As shown in Table 3, the proposed approach consistently achieves higher mIoU across all object categories. For example, the mIoU on guitar increases from 51.3% to 80.1%, and mug improves from 54.9% to 84.3%. The average mIoU reaches 56.4%, clearly surpassing the SADM average of 40.6%. These results show that the distilled dataset successfully captures the fine-grained geometric structure required for accurate part-level prediction.

Table 4: Comparison of dataset distillation performance across four variants of the ScanObjectNN benchmark.

| Variant | Random | Herding | K-Center | DM | SADM | Ours |
|---------|--------|---------|----------|-----|------|------|
| PB_T25 | 12.6 | 14.3 | 14.3 | 12.9 | 19.4 | **35.2** |
| PB_T25_R | 10.2 | 15.0 | 15.0 | 12.8 | 18.8 | **36.0** |
| PB_T50_R | 9.6 | 14.4 | 14.4 | 11.5 | 16.7 | **34.2** |
| PB_T50_RS | 13.5 | 15.1 | 15.1 | 13.7 | 17.6 | **32.6** |

**Evaluation on ScanObjectNN Benchmark Variants.**    To clearly assess the robustness of our proposed method under various challenging real-world scenarios, we conduct our experiments on the four variants of ScanObjectNN: PB_T25, PB_T25_R, PB_T50_R, and PB_T50_RS. As shown in Table 4, our method consistently outperforms the baselines across all variants, demonstrating robust performance regardless of the increasing difficulty of the datasets.

Table 5: Performance of the proposed adaptive shape morphing method with learnable weights compared with the static method of using fixed weights, evaluated on ScanObjectNN at PPC = 1.

| # of $L$ | 2 | 4 | 8 | 12 | 16 | 20 | 24 |
|---|---|---|---|---|---|---|---|
| Static | 19.8 | 30.1 | 30.8 | 32.5 | 31.7 | 31.4 | 30.8 |
| Adaptive | **21.0** | **32.6** | **32.4** | **34.8** | **35.1** | **35.6** | **33.0** |

Table 6: Effect of the proposed uniformity-aware matching loss using the penalty coefficient $\eta$.

| Datasets | ModelNet10 | | | ScanObjectNN | | |
|---|---|---|---|---|---|---|
| PPC | 1 | 3 | 10 | 1 | 3 | 10 |
| w/o $\eta$ | **88.4** | 88.3 | 90.1 | 30.7 | 40.6 | 47.6 |
| w/ $\eta$ | 87.7 | **89.8** | **92.2** | **32.6** | **41.3** | **49.8** |

## 4.3 QUALITATIVE RESULTS

Figure 3 illustrates how the resulting synthetic datasets evolve through optimization. For ModelNet10, the initial combined samples are generated by averaging the anchor samples with fixed weights and often appear as noisy point clouds lacking meaningful structure. In contrast, after applying the proposed method, the learnable weight vectors adaptively refine the combinations, producing structurally consistent 3D shapes. A similar trend is observed in the real-world dataset ScanObjectNN, where the initial combined samples, especially for classes such as door, sofa, and pillow, suffer from even more severe misalignment. Nevertheless, after optimization, the resulting samples exhibit significantly improved structural consistency. In both datasets, some combined samples appear as slight variations of anchor shapes, while others occasionally produce new structures not present in the anchors, demonstrating that the proposed method effectively balances structural preservation and shape diversity.

## 4.4 ABLATION STUDY

**Hyperparameter Selection.** We analyze the behavior of two key hyperparameters: the number of points per anchor sample ($N_2$) and the number of combined samples ($L$). Figure 4 (a) shows the accuracy for different values of $N_2$, while maintaining a fixed total memory budget by adjusting the number of anchors $M$ such that $MN_2 = N_1$, where $N_1$ is set to 1024. When evaluated with PointNet (Qi et al., 2017a) and PointNet++ (Qi et al., 2017b), PointNet performs better with smaller $N_2$ since it mainly focuses on global features and is less sensitive to the local structural variation of coarse anchors. In contrast, PointNet++ shows a sharp performance drop at $N_2 = 128$, indicating that it struggles to extract meaningful information when the resolution is too low. Based on this trade-off, we set $N_2 \approx 256$ within the budget for experiments in Tables 1 and 2.

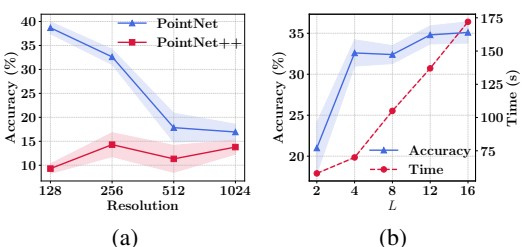

Figure 4: Analysis of performance on ScanObjectNN at PPC = 1 with respect to changes in hyperparameters. (a) Accuracy versus $N_2$ (number of points per anchor) under the same total memory budget. (b) Trade-off between classification accuracy and training time when varying $L$ (number of combined samples).

Figure 4 (b) also investigates the classification accuracy and training time in terms of the variation of the number of combined samples ($L$). The training time refers to the average time required to train the network during evaluation, averaged over 10 runs. In general, as $L$ increases, the accuracy is improved by enabling more expressive combinations, but the computational cost is also increased. The results show that, beyond $L = 4$, the accuracy almost saturates while the training time continues to grow. Based on this trade-off, we use $L = 4$ to strike a balance between the accuracy and efficiency, except using $L = 16$ for ModelNet10 (Wu et al., 2015) which has only 10 classes.

**Effectiveness of Learnable Shape Morphing.** To illustrate the contribution of the proposed learnable shape morphing strategy, we conducted two experiments. First, we compared the adaptive weighting scheme against a static baseline, where the weights are randomly initialized and remain fixed throughout the optimization. Table 5 reports the classification accuracy with varying the number of combined samples ($L$) from 2 to 24. The results show that the adaptive setting consistently

Table 7: Distillation results with various backbone architectures on ModelNet10 dataset with PPC set to 1.

| Datasets | SADM | | | | Ours | | | |
|---|---|---|---|---|---|---|---|---|
| Train/Test | PT | PC | PN++ | PN | PT | PC | PN++ | PN |
| PT | 35.6 | 17.1 | 12.4 | 24.7 | 34.2 | 24.3 | 12.7 | 59.6 |
| PC | 21.6 | 11.1 | 15.5 | 16.8 | 18.0 | 11.7 | 11.4 | 24.7 |
| PN++ | 44.2 | 21.8 | 11.5 | 33.8 | 35.4 | 23.6 | 23.4 | 78.4 |
| DG | 50.7 | 20.9 | 17.2 | 38.5 | **68.6** | 44.5 | 23.0 | 77.2 |
| PN | 49.0 | 20.5 | 25.9 | 35.9 | 57.0 | **51.6** | **55.4** | **87.7** |

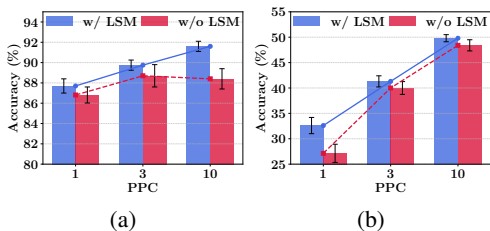

Figure 5: Ablation study evaluating the effectiveness of the proposed learnable shape morphing (LSM). (a) ModelNet10 and (b) ScanObjectNN.

outperforms the static setting. This indicates that learning the weights allows the model to control the relative contribution of each anchor sample more effectively, compensating for possible misalignments introduced by initial registration. Second, to evaluate the overall effect of the shape morphing strategy, we compared the framework with and without shape morphing, as shown in Figure 5. We see that applying the shape morphing improves the performance across all PPC settings. This validates that the shape morphing strategy enhances the diversity of synthetic dataset while generating semantically meaningful samples.

**Effectiveness of Uniformity-Aware Matching Loss.** To validate the effectiveness of the proposed uniformity-aware matching loss, we compared the models trained with and without using the penalty coefficient $\eta$ in (9). As shown in Table 6, the uniformity-aware matching loss with $\eta$ improves the performance across different PPC settings on ScanObjectNN and for higher PPC settings on ModelNet10. While a slight performance drop is observed at PPC = 1 on ModelNet10, the overall trend shows that applying the uniformity-aware matching loss leads to more stable and improved performance. In contrast, without using $\eta$, each partitioned subset $\mathcal{C}$ contributes equally to the overall loss regardless of how closely its spatial uniformity aligns with that of the original dataset, which can result in less reliable supervision. The observed performance gains suggest that the proposed uniformity-aware matching loss effectively mitigates the limitation of the subset partitioning.

**Results with Various Backbone Architectures.** We also distilled the synthetic datasets using Point Transformer (Zhao et al., 2021b), PointConv (Wu et al., 2019), PointNet++ (Qi et al., 2017b), PointNet (Qi et al., 2017a), and DGCNN (Wang et al., 2019), respectively, and evaluated them on ModelNet10. The results are summarized in Table 7. When a more complex backbone is used, it becomes inherently harder to align the feature distributions between the original and synthetic datasets. As more layers and operations such as local aggregation or attention are added, feature maps become more unstable, making it difficult to maintain a consistent alignment between the two distributions. In contrast, a simpler backbone produces more stable feature maps, so aligning the two distributions is easier. Therefore, we use PointNet as the backbone for all experiments.

## 5 CONCLUSION

In this paper, we first proposed a parameterization-based dataset distillation framework for 3D point clouds, capable of synthesizing informative and diverse samples under a constrained memory budget. To this end, we devised a learnable shape morphing strategy that diversifies synthetic samples by blending multiple anchor samples with coarser resolution in the initial synthetic set. Moreover, we designed a uniformity-aware matching loss that adaptively emphasizes the contribution of partitioned subsets of point clouds, improving the reliability of distribution matching between the original and synthetic datasets. Experimental results on five widely used benchmarks including ModelNet10 (Wu et al., 2015), ModelNet40 (Wu et al., 2015), ShapeNet (Chang et al., 2015), ScanObjectNN (Uy et al., 2019), and OmniObject3D (Wu et al., 2023) showed that the proposed method achieves substantial performance improvements over existing dataset distillation methods across various PPC settings.

## ACKNOWLEDGEMENTS

This work was supported in part by the National Research Foundation of Korea (NRF) grant funded by the [Ministry of Science and ICT (MSIT)] under Grant RS-2024-00392536, in part by the Institute of Information and Communications Technology Planning and Evaluation (IITP) grants funded by the Korean Government (MSIT), including the Leading Generative Artificial Intelligence (AI) Human Resources Development Program under Grant IITP-2025-RS-2024-00360227, in part by the Artificial Intelligence Graduate School Program [Ulsan National Institute of Science and Technology (UNIST)] under Grant RS-2020-II201336, and in part by the AI Star Fellowship Program (UNIST) under Grant RS-2025-25442824.

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

# APPENDIX

Large language models (LLMs) were used solely for language refinement, and all research content was generated entirely by the authors.

## A  ALGORITHM

Algorithm 1 outlines our parameterization-based dataset distillation method. The process begins by initializing multiple coarse anchors per class and aligning them via solving the assignment problem. During each distillation step, synthetic samples are generated through shape morphing, and the anchors and blending weights are optimized using a uniformity-aware distillation loss.

---

**Algorithm 1** Parameterization-Based Dataset Distillation via Learnable Shape Morphing

---

**Require:** Original dataset $\mathcal{D}_\mathrm{o}$, number of anchors $M$, number of combined samples $L$, size of synthetic dataset $S$

**Ensure:** Distilled anchors $\{\tilde{\mathbf{a}}_{i,m}\}$ and weights $\{\mathbf{w}_i^l\}$

1: Initialize $\mathcal{D}_\mathrm{init} = \{\{\mathbf{a}_{i,m}\}_{m=1}^M\}_{i=1}^S$ and $\mathcal{W} = \{\{\mathbf{w}_i^l\}_{l=1}^L\}_{i=1}^S$
2: Align the anchor samples within each group of $\mathcal{D}_\mathrm{init}$
3: **for** each distillation step **do**
4:     Construct synthetic dataset $\mathcal{D}_\mathrm{s} = \big\{\{\tilde{\mathbf{a}}_{i,m}\}_{m=1}^M \cup \{\sum_{m=1}^M w_{i,m}^l \cdot \tilde{\mathbf{a}}_{i,m}\}_{l=1}^L\big\}_{i=1}^S$
5:     Sample mini-batches $\mathcal{B}_\mathrm{o} \sim \mathcal{D}_\mathrm{o}$, $\mathcal{B}_\mathrm{s} \sim \mathcal{D}_\mathrm{s}$
6:     Partition $\mathcal{B}_\mathrm{o}$ into subsets $\mathcal{C}^1, \ldots, \mathcal{C}^M$
7:     Compute penalty coefficients $\eta^m = \exp\big(-\lambda(\nu(\mathcal{B}_\mathrm{o}) - \nu(\mathcal{C}^m))^2\big)$
8:     Compute distillation loss $\mathcal{L}_\mathrm{Distill} = \sum_{m=1}^M \eta^m \cdot \mathcal{L}_\mathrm{SADM}(\mathcal{C}^m, \mathcal{B}_\mathrm{s})$
9:     Update $\mathcal{D}_\mathrm{init}$, $\mathcal{W}$ w.r.t. $\mathcal{L}_\mathrm{Distill}$
10: **end for**

---

## B  EXPERIMENTAL DETAILS

### B.1  IMPLEMENTATION DETAILS

While the original $\mathcal{L}_\mathrm{SADM}$ consists of both $\mathcal{L}_\alpha$, which matches the entire feature map, and $\mathcal{L}_\beta$, which matches only the most prominent feature, we use only $\mathcal{L}_\alpha$ in our implementation. The configuration in Table 8(a) outlines the hyperparameters used for training the evaluation network. The network was optimized using stochastic gradient descent (SGD) with a learning rate of 0.01, a momentum of 0.9, and a weight decay of 0.0005. The batch size was set to 8, and training was conducted for 500 epochs. To adjust the learning rate during training, a StepLR scheduler was employed, with a step size of 250 and a decay factor of 0.1.

Table 8: (a) Hyperparameters used to train the evaluation network, and (b) hyperparameter settings of the baselines.

| Param | Value |
|-------|-------|
| Optimizer | SGD |
| Momentum | 0.9 |
| Weight Decay | 5e-4 |
| Batch Size | 8 |
| Learning Rate | 0.01 |
| Epochs | 500 |

(a)

| | DC | DM | MTT | PCC | SADM |
|---|-----|-----|-----|-----|------|
| Backbone | PointNet | PointNet | PointNet | PointNet | PointNet |
| Initialization | Random | Random | Random | Herding | Random |
| Batch Size $\mathcal{D}_\mathrm{o}$ | 8 | 8 | 8 | 8 | 8 |
| Batch Size $\mathcal{D}_\mathrm{s}$ | 8 | 8 | 8 | 8 | 8 |
| Learning Rate | 0.0001 | 1 | 0.0001 | 0.0001 | 10 |
| Distillation Steps | 2000 | 2000 | 2000 | 2000 | 2000 |

(b)

## B.2 Baselines

Since the official code for most baselines is either tailored for image-based tasks or unavailable, we re-implemented all baseline methods in our framework for a fair comparison. Table 8(b) summarizes the hyperparameter settings used for each baseline, including DC (Zhao et al., 2021a), DM (Zhao & Bilen, 2023), MTT (Cazenavette et al., 2022), PCC (Zhang et al., 2024), and SADM (Yim et al., 2025). All methods were implemented with PointNet as the backbone network and were trained under a consistent configuration where both the original and synthetic datasets used a batch size of 8, and each method was optimized for 2000 steps. For initialization, random initialization was used for most methods, except PCC, which employed herding initialization. While most methods used relatively low learning rates, DM and SADM adopted larger values of 1 and 10, respectively.

## C   Additional Experiments

### C.1   Plug-and-Play Application of the Proposed Method

To evaluate the independence of our method from specific distillation strategies, we apply it in a plug-and-play manner on top of DM. As shown in Table 9, our method substantially improves the performance of DM, particularly when PPC is low. For example, on ModelNet10 with PPC = 1, the accuracy rises from 25.8% to 79.8%, indicating that our method can effectively enhance even a weaker baseline. The last row (+Ours$^*$) presents the result of combining our method with SADM, which is identical to Table 1 in the main paper. This setting also shows the largest improvements at PPC 1. The consistent trend across both baselines suggests that our method is not tailored to any specific distillation framework but can serve as a general plug-and-play module that improves performance, especially under constrained memory budgets.

Table 9: Performance comparison with and without our plug-and-play method applied to DM and SADM. $*$ indicates the result of SADM combined with our method, which is identical to the performance already reported in the main paper.

| Dataset | ModelNet10 | | | ModelNet40 | | | ShapeNet | | | ScanObjectNN | | |
|---|---|---|---|---|---|---|---|---|---|---|---|---|
| PPC | 1 | 3 | 10 | 1 | 3 | 10 | 1 | 3 | 10 | 1 | 3 | 10 |
| DC | 32.8±8.5 | 74.5±2.6 | 84.6±0.6 | 50.3±2.0 | 66.0±1.1 | 74.3±0.9 | 48.7±1.6 | 56.6±1.1 | 63.7±0.8 | 15.2±2.0 | 24.6±2.2 | 38.5±1.6 |
| MTT | 27.8±5.8 | 73.6±1.7 | 85.3±1.2 | 33.4±2.1 | 59.5±0.6 | 73.4±0.5 | 32.4±2.6 | 53.5±2.0 | 62.3±1.1 | 14.3±2.5 | 20.1±1.3 | 37.1±2.0 |
| PCC | 33.0±8.0 | 70.7±1.6 | 86.3±1.1 | 55.3±1.4 | 66.2±1.6 | 77.9±0.9 | 50.9±3.5 | 58.9±1.7 | 65.4±0.8 | 16.0±2.4 | 25.5±2.2 | 34.6±1.4 |
| DM | 25.8±6.9 | 77.4±1.2 | 85.0±0.7 | 31.1±4.7 | 61.5±2.1 | 74.9±0.8 | 26.3±3.6 | 52.5±1.6 | 63.1±0.8 | 13.7±1.8 | 26.4±2.4 | 37.4±1.2 |
| + Ours | 79.8±1.6 | 82.4±1.4 | 86.4±1.4 | 55.6±0.1 | 67.3±0.6 | 76.3±0.6 | 52.3±1.4 | 59.9±1.3 | 63.5±0.4 | 18.1±0.8 | 29.3±0.1 | 37.3±1.0 |
| Δ | +54.0 | +5.0 | +1.4 | +24.5 | +5.8 | +1.4 | +26.0 | +7.4 | +0.4 | +4.4 | +2.9 | -0.1 |
| SADM | 35.9±8.2 | 83.5±0.7 | 87.4±1.1 | 54.8±1.3 | 71.3±0.7 | 79.6±0.6 | 51.1±2.3 | 62.2±1.6 | 68.0±0.5 | 17.6±1.5 | 32.6±1.6 | 43.7±2.0 |
| + Ours$^*$ | 87.7±0.7 | 89.8±0.5 | 92.2±0.5 | 73.2±1.1 | 80.3±0.5 | 82.5±0.6 | 60.5±1.1 | 65.9±0.6 | 68.9±0.6 | 32.6±1.6 | 41.3±1.1 | 49.8±0.7 |
| Δ | +51.8 | +6.3 | +4.8 | +18.4 | +9.0 | +2.9 | +9.4 | +3.7 | +0.9 | +15.0 | +8.7 | +6.1 |

### C.2   Ablation on Dataset Composition and Optimization Strategy

To analyze the effect of dataset composition and optimization strategy, we compare four synthetic dataset settings, each differing in how $\mathcal{D}_s$ is constructed and which parameters are optimized during distillation.

- $\mathcal{D}_s$ consists only of $\mathcal{D}_{comb}$, and only the combination weights $\mathcal{W}$ are optimized.

$$\mathcal{W}^* = \underset{\mathcal{W}}{\arg\min}\, \mathcal{L}_{Distill}(\mathcal{D}_o, \mathcal{D}_s) \quad \text{where} \quad \mathcal{D}_s = \mathcal{D}_{comb}. \tag{13}$$

- $\mathcal{D}_s$ includes both the fixed anchors $\mathcal{D}_{init}$ and the generated samples $\mathcal{D}_{comb}$, while only $\mathcal{W}$ is optimized.

$$\mathcal{W}^* = \underset{\mathcal{W}}{\arg\min}\, \mathcal{L}_{Distill}(\mathcal{D}_o, \mathcal{D}_s) \quad \text{where} \quad \mathcal{D}_s = \mathcal{D}_{init} \cup \mathcal{D}_{comb} \tag{14}$$

- $\mathcal{D}_s$ consists only of $\mathcal{D}_{comb}$, but both the anchors $\mathcal{D}_{init}$ and the weights $\mathcal{W}$ are optimized.

$$\{\mathcal{D}_{init}^*, \mathcal{W}^*\} = \underset{\{\mathcal{D}_{init}, \mathcal{W}\}}{\arg\min}\, \mathcal{L}_{Distill}(\mathcal{D}_o, \mathcal{D}_s) \quad \text{where} \quad \mathcal{D}_s = \mathcal{D}_{comb} \tag{15}$$

- $\mathcal{D}_s$ includes both $\mathcal{D}_{init}$ and $\mathcal{D}_{comb}$, and both are optimized during distillation.

$$\{\mathcal{D}^*_{init}, \mathcal{W}^*\} = \underset{\{\mathcal{D}_{init}, \mathcal{W}\}}{\arg\min} \mathcal{L}_{Distill}(\mathcal{D}_o, \mathcal{D}_s) \quad \text{where} \quad \mathcal{D}_s = \mathcal{D}_{init} \cup \mathcal{D}_{comb} \quad (16)$$

To ensure a fair comparison, we adjust the value of $L$ to equalize the total dataset size across all settings. As shown in Table 10, the results show that the best performance is achieved when both $\mathcal{D}_{init}$ and $\mathcal{W}$ are jointly optimized and both components are included in the final synthetic dataset.

Table 10: Ablation study on synthetic dataset composition and optimization strategy. Each row corresponds to a different formulation described in (13)–(16).

| Dataset | ModelNet10 | ModelNet40 | ShapeNet | ScanObjectNN |
|---------|------------|------------|----------|--------------|
| (13) | 80.4±0.9 | 54.5±0.4 | 46.3±2.1 | 18.5±0.8 |
| (14) | 78.8±1.1 | 55.5±0.9 | 51.3±0.7 | 18.1±1.3 |
| (15) | 87.3±0.9 | 71.9±0.9 | 59.3±0.8 | 28.4±1.2 |
| (16) | **87.7±0.7** | **73.2±1.1** | **60.5±1.1** | **32.6±1.6** |

## C.3 ABLATION ON DATA AUGMENTATION STRATEGY

Table 11: Ablation study comparing the baseline PointMixup and our method on ModelNet10 and ScanObjectNN under PPC = 1.

| Methods | ModelNet10 | ScanObjectNN |
|---------|------------|--------------|
| PointMixup | 82.3 | 21.9 |
| Ours | **87.7** | **32.6** |

To further investigate the effectiveness of our proposed method, we additionally conduct a comparison against PointMixup (Chen et al., 2020), a representative data augmentation technique for point clouds. Unlike PointMixup, which interpolates point clouds using a fixed coefficient without considering the distillation objective, our method synthesizes both the anchors and blending coefficients jointly with the distillation process. As shown in Table 11, our learnable shape morphing framework consistently outperforms PointMixup across both datasets.

## C.4 EFFECTIVENESS OF POINT CLOUD DATA AUGMENTATION

We evaluated the effect of standard point cloud augmentations on ScanObjectNN at PPC = 1, 3, and 10. The augmentation strategies include point jittering with Gaussian noise of standard deviation $\sigma = 0.001$, random scaling within the range 0.8 to 1.2, point dropping with a ratio of 0.875, and PointMixup Chen et al. (2020) with $\alpha = 0.2$. Methods that are not designed for point cloud (DM, DC, MTT) exhibited inconsistent behavior under these augmentations, with accuracy fluctuating depending on the PPC setting, suggesting that they do not reliably preserve structural information in point clouds. In contrast, point cloud dataset distillation methods (PCC, SADM, Ours) consistently benefited from the use of these augmentations. Furthermore, even when all methods were trained under the same augmented pipeline, our method achieved the highest accuracy at every PPC setting. These results demonstrate the robustness of proposed method to standard point cloud augmentations.

## D ADDITIONAL QUALITATIVE RESULTS

Figures 6 and 7 show additional qualitative results obtained with $L = 4$ on the ModelNet40 and ShapeNet datasets, respectively. These results demonstrate that our method generates slight variations from the original anchors. In all visualizations, blue point clouds represent the combined samples, while orange point clouds denote the anchors. Figures 8 and 9 present the distilled dataset under a storage budget of PPC = 3 on the ModelNet10 and ScanObjectNN datasets, respectively, where the increased capacity allows the synthesis of more diverse shapes. Figures 10 illustrate the results on the ModelNet10 datasets when $L = 16$, showing that our method can generate a wide range of shapes even from a limited set of anchors.

Table 12: Classification accuracy on ScanObjectNN under standard augmentations. "Aug." denotes whether augmentations were applied.

| PPC | Aug. | Random | Herding | K-Center | DM | DC | MTT | PCC | SADM | Ours |
|-----|------|--------|---------|----------|------|------|------|------|------|------|
| 1 | ✗ | 13.5 | 15.1 | 15.1 | 13.7 | 15.2 | 14.3 | 16.0 | 17.6 | **32.6** |
| | ✓ | 15.5 | 18.1 | 17.4 | 15.3 | 15.3 | 15.9 | 19.4 | 22.7 | **37.6** |
| 3 | ✗ | 19.7 | 26.9 | 18.8 | 26.4 | 24.6 | 20.1 | 25.5 | 32.6 | **41.3** |
| | ✓ | 19.2 | 29.0 | 21.4 | 21.1 | 22.6 | 19.6 | 31.7 | 35.1 | **44.2** |
| 10 | ✗ | 34.1 | 38.3 | 23.5 | 37.4 | 38.5 | 37.1 | 34.6 | 43.7 | **49.8** |
| | ✓ | 33.7 | 40.0 | 25.8 | 35.7 | 38.5 | 33.0 | 40.8 | 45.1 | **51.5** |

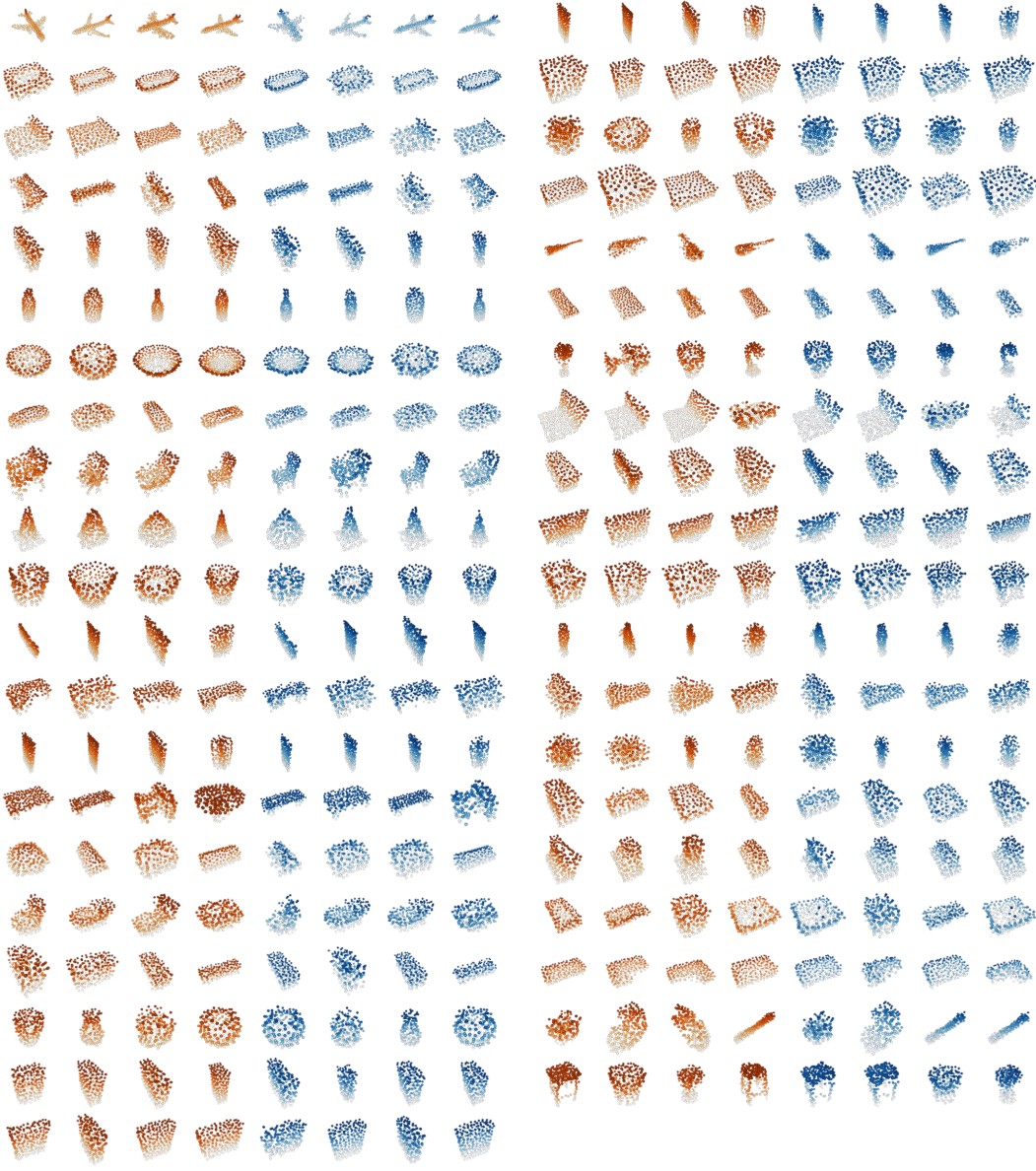

Figure 6: Visualization of distilled samples from ModelNet40 under a storage budget of PPC = 1.

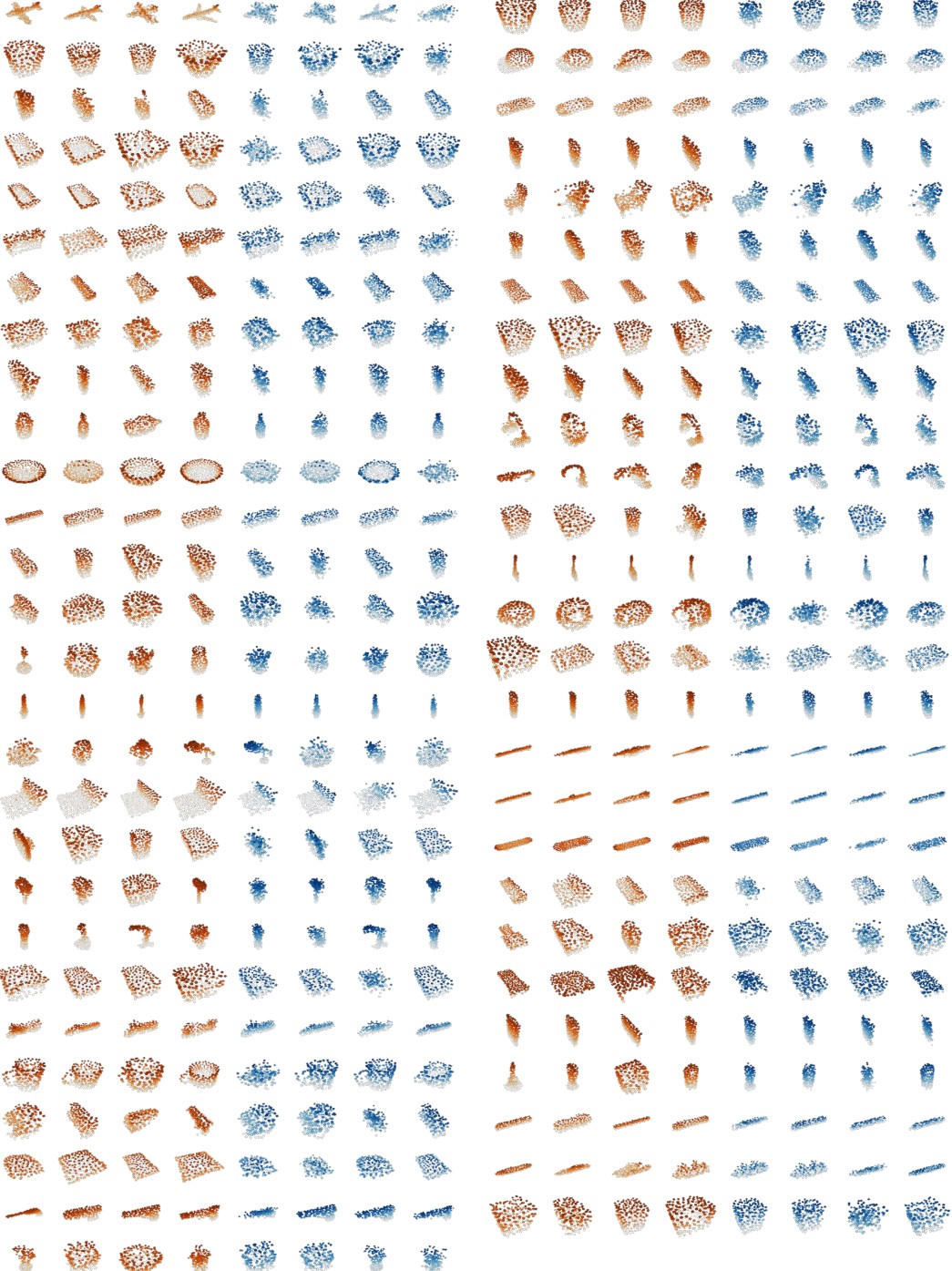

Figure 7: Visualization of distilled samples from ShapeNet under a storage budget of PPC = 1.

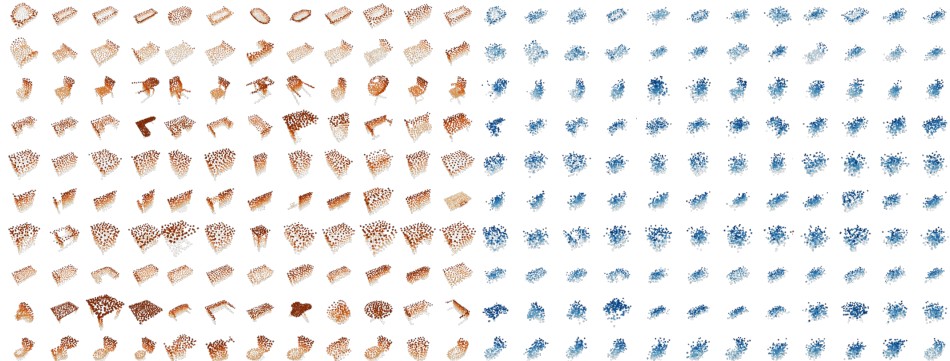

Figure 8: Visualization of distilled samples from ModelNet10 under a storage budget of PPC = 3.

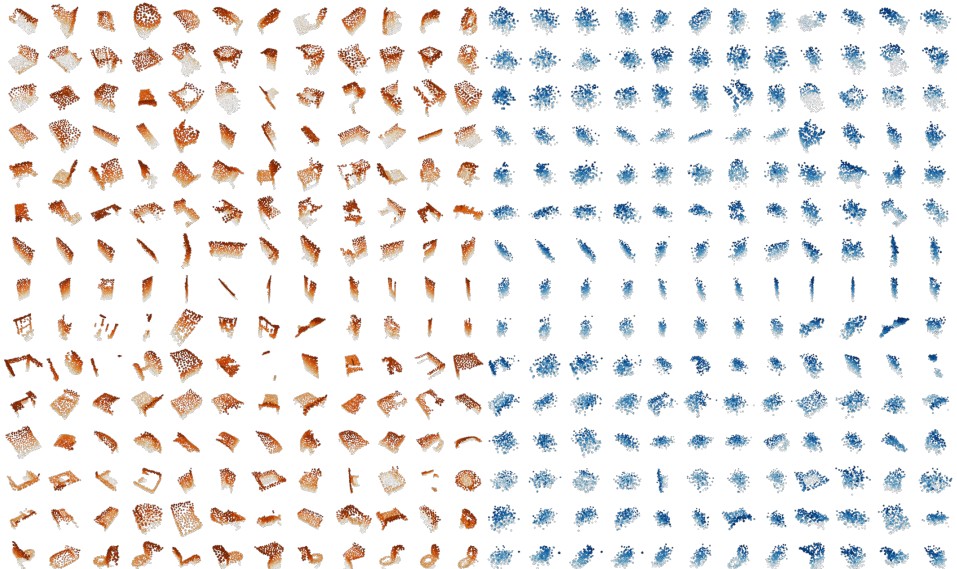

Figure 9: Visualization of distilled samples from ScanObjectNN under a storage budget of PPC = 3.

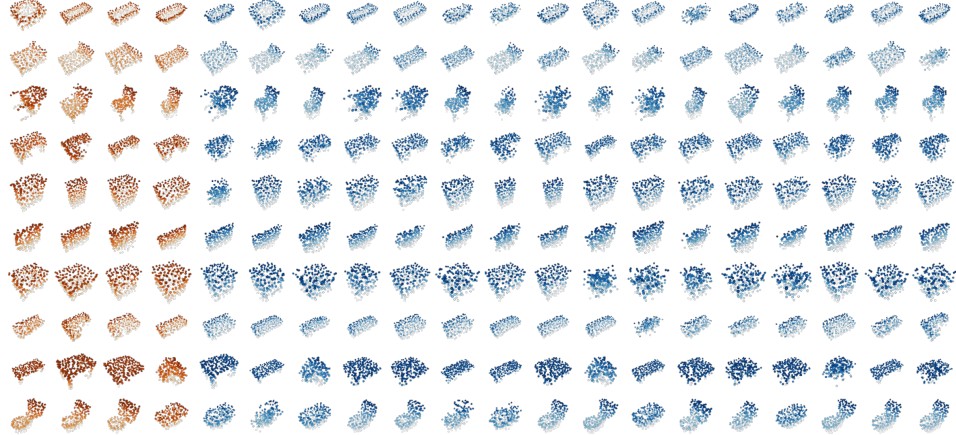

Figure 10: Visualization of synthetic samples from ModelNet10 with $L = 16$ under a storage budget of PPC = 1.

