# OpenReview forum: "Parameterization-Based Dataset Distillation of 3D Point Clouds through Learnable Shape Morphing"
_ICLR.cc/2026/Conference — ICLR 2026 Poster_

### Official Review · Reviewer_mBpX · 2025-10-28

**Soundness:** 3
**Presentation:** 3
**Contribution:** 3
**Rating:** 6
**Confidence:** 3

**Summary:**

This paper proposes a parameterized dataset distillation framework for 3D point clouds, which stores multiple coarse "anchor" point clouds for each synthetic sample and then generates additional synthetic samples through learnable shape deformation. To reduce the storage budget, the proposed method replaces the full-resolution synthetic sample with several low-resolution anchors plus lightweight blending weights. Experiments on ModelNet10/40, ShapeNet, and ScanObjectNN show substantial gains over prior DD and 3D-DD baselines, especially at low “points per class” (PPC) settings.

**Strengths:**

1. This paper proposes a novel 3D data parameterization method that expands the diversity of the synthetic dataset under a fixed budget. Replacing one full-res synthetic sample with M low-res anchors and L blended variants is reasonable.

2. The proposed method has strong empirical gains, especially at extreme compression ratios. For example, at PPC=1 with PointNet evaluation, the proposed method improves SOTA by large margins across ModelNet10/40, ShapeNet, and ScanObjectNN. Moreover, it also shows the best cross-architecture transfer in most cases.

3. The proposed method exhibits plug-and-play applicability. Applying it to DM and SADM has a significant performance boost.

**Weaknesses:**

1. The experiments are conducted on the classification task only. Although effective, there are still some other important tasks in 3D data, e.g., segmentation and detection. Can the proposed method be generalized to other tasks?

2. As the number of classes increases (e.g., ModelNet10 -> ModelNet40), the performance of the proposed method degenerates. Therefore, we may doubt whether the proposed method can be applied to large-scale datasets with more categories. It would be better if the author could provide more experimental results to validate its effectiveness.

3. The number of PPC is limited within \{1, 3, 10\}, which cannot achieve lossless performance in most datasets. More experimental results are required.

4. The source code is not provided. The reproducibility is not guaranteed.

**Questions:**

See weaknesses.

---

> ### Author Response · Authors · 2025-11-20
>
> We thank the reviewer for highlighting the novelty of proposed methods. We respond to the reviewer's comments as follows.
>
>
> **Q1. Generalization to other tasks**
>
> |PPC|Air|Bag|Cap|Car|Chair|Ear|Guitar|Knife|Lamp|Laptop|Motor|Mug|Pistol|Rocket|Skate|Table|Avg.|
> |-|-|-|-|-|-|-|-|-|-|-|-|-|-|-|-|-|-|
> |Whole|82.10|65.52|65.28|74.97|88.57|68.17|90.24|83.04|77.72|94.89|63.00|92.76|79.03|53.85|70.46|81.25|76.93|
> |Random|28.09|22.77|53.02|21.59|38.75|23.97|44.99|23.13|25.13|57.14|20.31|46.02|36.38|29.73|13.74|24.10|31.80|
> |Herding|31.67|36.11|47.12|22.07|46.37|34.85|50.36|54.23|20.89|65.24|16.25|54.94|34.96|30.75|31.45|42.32|38.72|
> |KCenter|31.67|36.11|47.12|22.07|46.37|34.85|50.36|54.23|20.89|65.24|16.25|54.94|34.96|30.75|31.45|42.32|38.72|
> |SADM|29.90|30.60|52.34|21.55|49.37|23.12|51.32|65.97|29.34|68.84|14.97|51.70|39.81|35.35|38.51|46.50|40.57|
> |Ours|51.45|51.80|59.03|36.54|70.78|42.61|80.13|76.55|31.83|81.00|26.44|84.29|61.76|40.47|49.18|59.09|56.43|
>
> The detection task cannot be tested in the proposed method, since it is defined on 3D point cloud scenes that usually contains multiple objects as well as background, whereas the proposed method is designed for a single 3D object. However, as the reviewer suggested, we additionally evaluated the performance of the proposed method applied to a downstream task of part segmentation on ShapeNet using PointNet segmentation architecture. The results are shown in the table above, where we observe that the proposed method significantly outperforms baseline coreset selection and distillation methods across all object categories. The revised version of the paper includes these results in Table 3 with the corresponding discussion.
>
> **Q2. Scalability to large-scale datasets**
>
> Thanks for this insigtful comment. An increase in the number of classes alone is not a decisive factor of performance. The achievable accuracy depends on how difficult it is to compress the original dataset into a small number of synthetic samples. This difficulty is influenced by the number of classes as well as the intra-class variation, presence of noise or occlusion, etc. For example, ModelNet10, ModelNet40, and ShapeNet contain 10, 40, and 55 class categories, respectively, and our distilled dataset at PPC=10 reaches 100 percent, 93 percent, and 84 percent of the full-dataset accuracy. On the other hand, ScanObjectNN contains only 15 classes, yet its relative performance at PPC=10 is about 79 percent. This is because ScanObjectNN exhibits real-world noise, occlusion, and clutter, making dataset distillation more challenging.
>
> |PPC | Random | Herding | K-Center | DM | DC | MTT | PCC | SADM | Ours  | Whole|
> |-|-|-|-|-|-|-|-|-|-|-|
> |1  | 24.0 | 30.5 | 30.5 | 15.1 | 31.1 | 25.8 | 35.8 | 33.2 | 41.9|
> |3  | 40.9 | 42.3 | 35.9 | 40.3 | 43.6 | 44.8 | 45.0 | 51.6 | 58.4| 74.98|
> |10 | 59.5 | 59.3 | 58.3 | 60.0 | 56.2 | OOM | 57.0 | 65.2 | 68.4|
>
> As the reviewer suggested, to further examine the scalability of the proposed method, we also evaluated our method on OmniObject3D, a large-scale dataset with over 200 class categories. We used 80 percents of the data for distillation and the remaining 20 percents for testing, respectively. We ensured that each class contained at least four test samples, resulting in 156 classes being used in our experiments. As shown in the table below, despite significantly larger number of classes, our distilled dataset yields around 90 percents of the full-data performance at PPC=10, and furthermore consistently achieves the highest accuracy outperforming all existing dataset distillation baselines. We have included these results in Table 1, along with the corresponding discussion, in the revised manuscript.
>
> **Q3. Performance at high PPC.**
>
> Following the reviewer’s suggestion, we additionally conducted experiments to evaluate the proposed method at larger PPC settings, using PPC = 30 for ModelNet40 and PPC = 50 for ScanObjectNN. As shown in the table below, our method continues to outperform all baselines on both datasets even under higher PPCs. Notably, PPC = 30 for ModelNet40 corresponds to approximately 12\% of the full dataset, while PPC = 50 for ScanObjectNN corresponds to only about 4.5\% of the full dataset. Despite using such small portions of each dataset, the distilled datasets achieve performance close to the full-dataset results on both ModelNet40 and ScanObjectNN, demonstrating the reliability of the proposed method across various PPC settings.
>
> |PPC|Dataset | Random | Herding | K-Center | DM | SADM | Ours  | Whole|
> |-|-|-|-|-|-|-|-|-|
> |30|ModelNet40|82.5|84.6|77.7|82.4|84.8|86.3|88.8|
> |50|ScanObjectNN|51.1|52.1|36.9|50.2|55.1|59.1|63.4|
>
>
>
> **Q4. Reproducibility**
>
> We provided all hyperparameter settings and implementation details needed to reproduce the results in the paper and supplementary material. The official code will be made publicly available after publication.

---

### Official Review · Reviewer_3htV · 2025-10-29

**Soundness:** 3
**Presentation:** 3
**Contribution:** 2
**Rating:** 4
**Confidence:** 2

**Summary:**

This paper introduces the first parameterization-based dataset distillation (DDP) framework for 3D point clouds, enabling more diverse synthetic samples under fixed memory budgets. Rather than optimizing full-resolution synthetics (as in prior DD), it initializes with M coarse anchors per class (MN₂ ≤ N₁) and generates L additional samples via learnable convex morphing of aligned anchors (akin to 3D mixup). Optimization uses a uniformity-aware SADM loss, which partitions full-res originals into coarse subsets and downweights non-uniform partitions via CV-based penalties. Evaluated on ModelNet10/40, ShapeNet, and ScanObjectNN with PointNet (500 epochs), it achieves SOTA DD accuracy (e.g., 82.5% MN40@10PPC vs. prior 79.6% SOTA) and strong cross-arch transfer (e.g., 47.7% PN++ on MN40@1PPC). However, large gaps to oracle ("Whole") persist (e.g., 73% vs 89% MN40@1PPC).

**Strengths:**

1. First to parameterize point clouds via coarse anchors + morphing, yielding ~M+L samples vs. 1 prior—simple, zero-extra-cost diversity boost (storage: 96MN₂KC + 32L(M-1)KC bits ≤ prior budget).
2. Consistent SOTA over 2D-transferred (DM/DC/MTT), 3D-tailored (PCC/SADM), and coresets across 4 datasets/PPCs; excellent cross-arch gen (Table 2) shows robustness.
3. Anchor alignment (Hungarian on dist matrices) + uniformity penalty elegantly handles irregularity/coarseness; ablation potential high (e.g., morphing ablation implied).
4. Fig.1/2 crisp; budget math (Sec.3.4) rigorous; oracle baselines throughout.

**Weaknesses:**

1. Morphing is point-wise linear interpolation post-alignmen fundamentally 3D mixup on anchors. Lacks novelty over image DDP (IDC/FreD/HaBa) or point cloud mixup; alignment brittle for topology-varying shapes (e.g., chair/table).
2. DD perf drops ~15-40% vs. oracle (e.g., 32% vs 63% ScanObj@1PPC), understates distillation limits; no segmentation/transfer tasks (e.g., PartNet).
3. Baseline fairness is unclear. Do priors (PCC/SADM) use data aug (jitter/rotate/scale/chroma std for PointNet)? Paper implies yes ("standard"), but unconfirmed; oracle likely uses aug, inflating gaps.
4. Limited scope/ablation: No hyperparam sensitivity (M/L/N₂); fixed PointNet (not DGCNN/PT from start); no real-time distilling or downstream (e.g., detection).

**Questions:**

1. How does learnable morphing outperform fixed-uniform weights or point cloud mixup (e.g., PointMixup) baselines? Ablate alignment (e.g., w/o Hungarian)?
2.Explicitly confirm aug for all methods/oracle (e.g., Table: jitter σ=0.01, rot [±15°], scale [0.8-1.2])—did SADM/PCC reimpls match your 500-epoch setup?
3. Add "Oracle (Full Dataset)" column w/ aug details; report relative gap (e.g., 82% of oracle @10PPC)? Test on downstream (PartNet seg, KITTI det)?
4. Ablate M/L/N₂ tradeoffs (Fig?); why N₂=256, M=4, L=3? Test higher-res anchors or topology-aware alignment (e.g., functional maps)?
5. Show qualitative synthetics (morphing fails?); robustness to noisy ScanObj aug (occlusion/clutter)? Extend to dynamic scenes (nuScenes LiDAR)?

---

> ### Author Response · Authors · 2025-11-20
> **Response for Reviewer 3htV(1/3)**
>
> We thank the reviewer to recognize the originality of our paper. We respond to the reviewer's comments as follows.
>
> **Q1. Novelty beyond 3D Mixup and existing image DDP methods.**
>
> Thanks for this insightful comment. We would like to note that existing distilled dataset parameterization (DDP) methods for images cannot be straightforwardly applied to 3D point clouds due to unordered nature of point structures. As recognized by reviewers, our main novelty lies on the first formulation of the DDP problem for 3D point clouds and its solution via learnable shape morphing.
>
> |Methods | ModelNet10 | ScanObjectNN |
> |---|---|---|
> |PointMixup | 82.3 | 21.9|
> |Ours | **87.7** | **32.6**|
>
> Moreover, although our framework and PointMixup both generate intermediate shapes between existing samples, the underlying mechanisms are fundamentally different. Mixup is a data augmentation technique that generates new samples by interpolating two point clouds using a fixed coefficient, without considering the objectives of dataset distillation. In contrast, our method blend multiple anchors using learnable coefficients that are optimized jointly with the distillation objective. As a result, both the anchors and the morphed samples are synthesized in a way that directly minimizes the distribution matching loss, enabling the synthetic dataset to capture distillation-relevant structure. As the reviewer suggested, we compared the performance at PPC=1 between PointMixup and ours in the table above, which shows that our learnable shape morphing consistently outperforms PointMixup. We have included these results and corresponding discussion in Supplementary C.3.
>
> **Q2. Large performance drop from oracle**
>
> | Dataset       |  |      ModelNet10     |           |  |     ModelNet40      |           |   |     ShapeNet      |           | |     ScanObjectNN      |           |
> |:---------------:|:-------------:|:-------------:|:-------------:|:-------------:|:-------------:|:-------------:|:-------------:|:-------------:|:-------------:|:-------------:|:-------------:|:-------------:|
> |     PPC      | 1       | 3     | 10    | 1       | 3     | 10    | 1      | 3     | 10    | 1        | 3     | 10     |
> | Ratio         | 0.25        | 0.75      | 2.5       | 0.4         | 1.2       | 4.0       | 0.15       | 0.45      | 1.5       | 0.15         | 0.45      | 1.5        |
> | Acc.          | 87.7        | 89.8      | 92.2      | 73.2        | 80.3      | 82.5      | 60.5       | 65.9      | 68.9      | 32.6         | 41.3      | 49.8       |
> | Rel.          | 95.1        | 97.4      | 100.0     | 82.4        | 90.4      | 92.9      | 73.3       | 79.9      | 83.5      | 51.4         | 65.1      | 78.5       |
> | Whole         |        | 92.2      |       |        | 88.8      |      |       | 82.5      |      |         | 63.4      |       |
>
> Large performance drop at PPC=1 is an expected natural consequence. The above table reports the ratios of used storage budget at PPCs with respect to the whole dataset (**Ratio**), and the relative performance (**Rel.**) of the accuracy (**Acc.**) compared to that of the whole dataset. In ScanObjectNN, PPC=1 means that the network is trained by using only 0.15 percent of the storage space compared to the full dataset, yielding unavoidable performance drop at an extremely high compression ratio. However, it is worth to note that the performance of the proposed method is significantly much higher than that of existing alternatives of coreset selection and dataset distillation (e.g., 32.6 (ours) vs. 17.6 (existing SOTA) at PPC=1 on ScanObjectNN), as shown in Table 1 in the paper.
>
> **Q3. Baseline fairness and data augmentation**
>
> We confirm that no data augmentation was used for any method including the oracle. All baselines (DM, DC, MTT, PCC, SADM) were fully re-implemented in our framework and trained under the exact same 500-epoch configuration. Therefore, the reported results of all methods are evaluated under the identical and augmentation-free setting, ensuring a strictly fair comparison. This is clarified in Section 4.1 (**Implementation Details**) of the revised manuscript.

---

> > ### Author Response · Authors · 2025-11-20
> > **Response for Reviewer 3htV(2/3)**
> >
> > **Q4. More ablation study**
> >
> > **Hyperparameter sensitivity.**  We politely mention that the ablation study of hyperparameter selection was already included in Figure 4, and the results were discussed in Section 4.4 (**Hyperparameter Selection**).
> > Note that $M$ and $N_2$ are tightly coupled through the memory constraint $MN_2 \\le N_2$, meaning that increasing one necessarily reduces the other. For this reason, we analyze the trade-offs of $M$ and $N_2$ jointly under the fixed budget. As shown in Figure 4(a), $N_2=256$ with $M=4$ produces balanced results across both architectures of PointNet and PointNet++, offering sufficient resolution while preserving anchor diversity.
> > In contrast, $L$ occupies only a negligible portion of the memory budget and is therefore largely independent of the $M$ and $N_2$ configuration. As shown in Figure 4(b), the accuracy improves as $L$ increases, but the computational cost also grows rapidly. Beyond $L=4$, accuracy gains saturate while the training time substantially increases. Therefore, for the main experiments we adopt $L=4$, which provides a strong balance between accuracy and efficiency, while using $L = 16$ only for ModelNet10 due to its smaller number of classes.
> >
> > **Effect of different architectures.**  We already included the experimental results with different backbone architectures in Table 5 and discussed the results in Section 4.4 (**Results with Various Backbone Architectures**).
> >
> > | Distillation | Methods | PT   | PC   | PN++ | PN   | Avg.  |
> > |-|-|-|-|-|------|-------|
> > | **PT** | SADM    | **35.6** | 17.1 | 12.4 | 24.7 | 22.5 |
> > || Ours    | 34.2 | **24.3** | **12.7** | **59.6** | **32.7** |
> > | **PC**| SADM    | **21.6** | 11.1 | **15.5** | 16.8 | 16.3 |
> > || Ours    | 18.0 | **11.7** | 11.4 | **24.7** | **16.5** |
> > | **PN++**    | SADM    | **44.2** | 21.8 | 11.5 | 33.8 | 27.8 |
> > || Ours    | 35.4 | **23.6** | **23.4** | **78.4** | **40.2** |
> > | **PN**        | SADM    | 49.0 | 20.5 | 25.9 | 35.9 | 32.8 |
> > |              | Ours    | **57.0** | **51.6** | **55.4** | **87.7** | **62.9** |
> > | **DG**        | SADM    | 50.7 | 20.9 | 17.2 | 38.5 | 31.8 |
> > |              | Ours    | **68.6** | **44.5** | **23.0** | **77.2** | **53.3** |
> >
> > We also provide additional experimental results using various backbone architectures for distillation, evaluated on the ModelNet10 dataset, comparing the SOTA SADM baseline with the proposed method in the table above. Our method with the PointNet backbone consistently achieves the highest performance among diverse variations, demonstrating better generalization performance.
> >
> > **Effect of learnable morphing.**  We already included the ablation study of learnable morphing in Table 5, and discussed the results in Section 4.4 (**Effectiveness of Learnable Shape Morphing**). Table 5 shows that the adaptive weighting scheme consistently outperforms the static setting. This indicates that learning the weights allows the model to control the relative contribution of each anchor sample more effectively, compensating for possible misalignments introduced by initial registration.
> >
> > **Effect of alignment (e.g., w/o Hungarian)**  As the reviewer suggested, we conducted comparative experiments where the anchors are constructed with and without Hungarian matching to evaluate the effect of the alignment step. As shown in the table below, without alignment, shape morphing becomes less stable leading to consistent performance drops on both ModelNet10 and ScanObjectNN.
> >
> > |Datasets |  |  ModelNet10  |  |  | ScanObjectNN  |  |
> > |:-:|:-:|:-:|:-:|:-:|:-:|:-:|
> > |PPC | 1 | 3 | 10 |  1 | 3 | 10|
> > |w/o Align | 86.6 | 89.6 | 91.9 | 31.6 | 41.3 | 49.4|
> > |w/ Align  | 87.7 | 89.8 | 92.2 | 32.6 | 41.3 | 49.8|

---

> ### Author Response · Authors · 2025-11-20
> **Response for Reviewer 3htV(3/3)**
>
> **Q5. Test on downstream task**
>
> Note that the proposed point cloud distillation methods are designed for a single object, where each sample represents a distinct class label. On the other hand, a point cloud scene usually contains multiple objects and background, which does not exhibit consistent structure across different scenes. Therefore, dataset distillation and downstream tasks such as KITTI det and nuScenes LiDAR are beyond the scope of this work. However, as the reviewer suggested, we evaluated the performance of a downstream task of part segmentation on ShapeNet using PointNet segmentation architecture. The results are shown in the table below, where we observe that the proposed method significantly outperforms baseline coreset selection and distillation methods across all object categories. The revised version of the paper includes these results in Table 3 with the corresponding discussion.
>
> |PPC | Air | Bag | Cap | Car | Chair | Ear | Guitar | Knife | Lamp | Laptop | Motor | Mug | Pistol | Rocket | Skate | Table | Avg. |
> |-|-|-|-|-|-|-|-|-|-|-|-|-|-|-|-|-|-|
> |Whole | 82.10 | 65.52 | 65.28 | 74.97 | 88.57 | 68.17 | 90.24 | 83.04 | 77.72 | 94.89 | 63.00 | 92.76 | 79.03 | 53.85 | 70.46 | 81.25 | 76.93|
> |Random | 28.09 | 22.77 | 53.02 | 21.59 | 38.75 | 23.97 | 44.99 | 23.13 | 25.13 | 57.14 | 20.31 | 46.02 | 36.38 | 29.73 | 13.74 | 24.10 | 31.80 |
> |Herding | 31.67 | 36.11 | 47.12 | 22.07 | 46.37 | 34.85 | 50.36 | 54.23 | 20.89 | 65.24 | 16.25 | 54.94 | 34.96 | 30.75 | 31.45 | 42.32 | 38.72 |
> |K Center | 31.67 | 36.11 | 47.12 | 22.07 | 46.37 | 34.85 | 50.36 | 54.23 | 20.89 | 65.24 | 16.25 | 54.94 | 34.96 | 30.75 | 31.45 | 42.32 | 38.72 |
> |SADM | 29.90 | 30.60 | 52.34 | 21.55 | 49.37 | 23.12 | 51.32 | 65.97 | 29.34 | 68.84 | 14.97 | 51.70 | 39.81 | 35.35 | 38.51 | 46.50 | 40.57 |
> |Ours | 51.45 | 51.80 | 59.03 | 36.54 | 70.78 | 42.61 | 80.13 | 76.55 | 31.83 | 81.00 | 26.44 | 84.29 | 61.76 | 40.47 | 49.18 | 59.09 | 56.43 |
>
> **Q6. Qualitative results of distilled samples**
>
> We already visualized the resulting distilled samples in Figure 3 in the main paper and Figures 6$\sim$9 in the supplementary material. Note that our purpose is not to generate visually pleasing distilled samples but optimal ones for training, which means that noisy or irregular appearances do not always indicate failure cases.
>
> **Q7. Robustness to noisy ScanObjectNN**
> |Dataset | Random | Herding | K-Center | DM | SADM | Ours|
> |---|---|---|---|---|---|---|
> |PB\_T25    | 12.6 | 14.3 | 14.3 | 12.9 |19.4  |35.2|
> |PB\_T25\_R | 10.2 | 15.0 | 15.0 | 12.8 |18.8  |36.0|
> |PB\_T50\_R | 9.6 | 14.4 | 14.4 | 11.5 |16.7   |34.2|
> |PB\_T50\_RS| 13.5 | 15.1 | 15.1 | 13.7 |17.6  |32.6|
>
> We would like to clarify that our main experiments on ScanObjectNN are conducted using the **PB\_T50\_RS** split, which is the hardest variant of the dataset. Since our method is evaluated directly on this variant, the reported performance already reflects its robustness to the noisy conditions. To further analyze the performance of the proposed method, we conducted experiments on three additional variants of ScanObjectNN. The results show that, regardless of the difficulty level, the proposed method consistently shows strong performance.

---

> > ### Comment · Reviewer_3htV · 2025-11-26
> >
> > Thank you for confirming that the compared methods do not use data augmentation.
> >
> > But it would be great to show all results with the common data augmentation, like point jittering, point drop, mixup etc.
> >
> > It is because the data augmentation is a standard practice to train the point cloud networks.
> >
> > My concern is that the data augmentation will hinder the improvements of the proposed method (which somehow uses weighted mix-up).

---

> > > ### Author Response · Authors · 2025-11-30
> > >
> > > We appreciate the reviewer’s suggestion to report results with standard data augmentation. In response, we conducted additional experiments on ScanObjectNN at PPC settings of 1, 3, and 10 using widely adopted augmentations (jitter: $\sigma=0.001$, scale: [0.8–1.2], point drop: 0.875, PointMixup: $\alpha=0.2$). For methods that are not designed for point cloud dataset distillation (DM, DC, MTT), applying standard point cloud augmentations produces inconsistent results, with performance fluctuating across PPC settings. This indicates that they fail to preserve faithful structural patterns in point clouds. In contrast, dataset distillation approaches for point clouds (PCC, SADM, Ours) show clear improvements when standard augmentations are applied. Moreover, even when all methods are trained with identical augmentations, our method still achieves the highest accuracy at every PPC setting. These observations indicate that standard point cloud augmentations do not hinder the performance gains of our method. We included this results in Supplementary C.4.
> > >
> > > |PPC|Aug.|Random|Herding|K-Center|DM|DC|MTT|PCC|SADM|Ours|
> > > |-|-|-|-|-|-|-|-|-|-|-|
> > > |1|w/o Aug.|13.5|15.1|15.1|13.7|15.2|14.3|16.0|17.6|32.6|
> > > |1|w/ Aug.|15.5|18.1|17.4|15.3|15.3|15.9|19.4|22.7|37.6|
> > > |3|w/o Aug.|19.7|26.9|18.8|26.4|24.6|20.1|25.5|32.6|41.3|
> > > |3|w/ Aug.|19.2|29.0|21.4|21.1|22.6|19.6|31.7|35.1|44.2|
> > > |10|w/o Aug.|34.1|38.3|23.5|37.4|38.5|37.1|34.6|43.7|49.8|
> > > |10|w/ Aug.|33.7|40.0|25.8|35.7|38.5|33.0|40.8|45.1|51.5|

---

### Official Review · Reviewer_tN6U · 2025-11-01

**Soundness:** 3
**Presentation:** 3
**Contribution:** 3
**Rating:** 6
**Confidence:** 4

**Summary:**

This paper presents the first parameterization-based dataset distillation framework for 3D point clouds, enabling the creation of compact synthetic datasets under strict memory constraints. Unlike prior 2D methods, it introduces learnable shape morphing to generate diverse samples by blending coarser anchor shapes with learnable weights, maximizing diversity within the same memory budget. A uniformity-aware matching loss further enforces structural consistency by balancing contributions from different data partitions. Joint optimization of anchors and morphing weights minimizes this loss, yielding superior results on ModelNet10/40, ShapeNet, and ScanObjectNN, especially under tight memory conditions.

**Strengths:**

+ The paper introduces a novel parameterization-based dataset distillation framework for 3D point clouds
+ Outperforms existing dataset distillation techniques across all four standard benchmarks (ModelNet10, ModelNet40, ShapeNet, and ScanObjectNN) at various Point Clouds Per Class (PPC) settings.
+ On ModelNet10 at PPC=1, it achieves an accuracy of 87.7% (previous SOTA 35.9%).
+ On the challenging real-world dataset, ScanObjectNN improves the previous SOTA from 17.6% to 32.6% at PPC=1.

**Weaknesses:**

- Comparison of cross-architecture generalization performance at PPC = 1 (Table 2) is made on relatively older models (PointNet++, PointConv, and PointTransformer) and not on recent models (e.g., PointMamba). Hence, it is unclear if the proposed approach works on recent methods.
- Results limited to a few selected datasets: the paper does not mention if the results of the real-world dataset ScanObjectNN [1] presented in the paper are for the hardest variant PB_T50_RS.
- Moreover, these datasets have already reached saturation points; recent real-world datasets (e.g., 3DGrocery100[2], OmniObject3D[3]) are not used to evaluate the proposed approach.

[1] Uy, Mikaela Angelina, et al. "Revisiting point cloud classification: A new benchmark dataset and classification model on real-world data." Proceedings of the IEEE/CVF international conference on computer vision. 2019.

[2] Sheshappanavar, Shivanand Venkanna, et al. "A benchmark grocery dataset of real-world point clouds from single view." 2024 International Conference on 3D Vision (3DV). IEEE, 2024.

[3] Wu, Tong, et al. "Omniobject3d: Large-vocabulary 3d object dataset for realistic perception, reconstruction and generation." Proceedings of the IEEE/CVF Conference on Computer Vision and Pattern Recognition. 2023.

**Questions:**

N/A

---

> ### Author Response · Authors · 2025-11-20
>
> We thank the reviewer to recognize the novelty of our paper. We respond to the reviewer's comments as follows.
>
> **Q1. Experimental results on recent model (PointMamba)**
>
> |Method | ModelNet10 | ModelNet40 | ShapeNet | ScanObjectNN|
> |---|---|---|---|---|
> |Random | 29.2 | 34.1 | 17.4 | 18.9|
> |DC     | 20.7 | 24.3 | 13.1 | 17.7|
> |DM     | 14.0 | 12.3 | 5.6 | 14.8|
> |MTT    | 31.6 | 33.4 | 17.3 | 18.9|
> |PCC    | 28.9 | 38.3 | 30.7 | 13.6|
> |SADM   | 28.4 | 35.9 | 25.6 | 13.2|
> |Ours   | **69.4** | **59.6** | **49.0** | **19.7** |
>
> Thanks for this constructive comment. As the reviewer suggested, we provide the cross-architecture generalization performance evaluated on a more recent model of PointMamba. As shown in the table above, the proposed method consistently far surpasses all existing distillation baselines on this recent model as well. In the revised manuscript, these results have been included in Table 2 and Section 4.2 (**Cross-Architecture Generalization.**)
>
> **Q2. Clarification of the ScanObjectNN variant**
>
> |Dataset | Random | Herding | K-Center | DM | SADM | Ours|
> |---|---|---|---|---|---|---|
> |PB\_T25    | 12.6 | 14.3 | 14.3 | 12.9 |19.4  |35.2|
> |PB\_T25\_R | 10.2 | 15.0 | 15.0 | 12.8 |18.8  |36.0|
> |PB\_T50\_R | 9.6 | 14.4 | 14.4 | 11.5 |16.7   |34.2|
> |PB\_T50\_RS| 13.5 | 15.1 | 15.1 | 13.7 |17.6  |32.6|
>
> All our experiments were conducted using the PB\_T50\_RS variant, which is the hardest setting of ScanObjectNN. In addition, we conduct the experiments on other variants of ScanObjectNN. To further analyze the performance of the proposed method, we conducted experiments on three additional variants of ScanObjectNN. The results show that, regardless of the difficulty level, the proposed method consistently shows strong performance. We have included this results in the Table 4 of revised manuscript. Please see the Section 4.2 (**Evaluation on ScanObjectNN Benchmark Variants.**)
>
> **Q3. Additional experiments on recent real-world datasets**
>
> |PPC | Random | Herding | K-Center | DM | DC | MTT | PCC | SADM | Ours  | Whole|
> |---|---|---|---|---|---|---|---|---|---|---|
> |1  | 24.0 | 30.5 | 30.5 | 15.1 | 31.1 | 25.8 | 35.8 | 33.2 | 41.9|
> |3  | 40.9 | 42.3 | 35.9 | 40.3 | 43.6 | 44.8 | 45.0 | 51.6 | 58.4| 74.98|
> |10 | 59.5 | 59.3 | 58.3 | 60.0 | 56.2 | OOM | 57.0 | 65.2 | 68.4|
>
> Following the reviewer’s suggestion, we additionally evaluated the performance of our method on a recent real-world dataset of OmniObject3D. It does not provide an official train–test split, and therefore we used 80 percents of the data for distillation and the remaining 20 percents for testing, respectively. We ensured that each class contained at least four test samples, resulting in 156 classes being used in our experiments. As shown in the table above, our method consistently achieves the highest accuracy, outperforming all existing dataset distillation baselines including the SOTA method of SADM. We have incorporated these results into Table 1 and added the corresponding discussion in Section 4.2 (**Evaluation on PointNet.**) in the revised manuscript.

---

### Official Review · Reviewer_5FmN · 2025-11-03

**Soundness:** 2
**Presentation:** 2
**Contribution:** 3
**Rating:** 4
**Confidence:** 4

**Summary:**

This paper introduces a novel parameterization-based dataset distillation framework for 3D point clouds. The approach leverages learnable shape morphing to generate diverse synthetic samples within a constrained memory budget. The core idea involves using multiple coarser anchor samples and blending their shapes with learnable weights.  While the paper presents promising results on standard benchmarks, several critical issues need to be addressed before it can be considered for acceptance.

**Strengths:**

1. The application of parameterization techniques to dataset distillation for 3D point clouds is a relatively unexplored area, and this work introduces a potentially efficient approach. The idea of using learnable shape morphing to generate diverse synthetic samples is innovative.
2. The paper demonstrates promising performance on several benchmarks (ModelNet10, ModelNet40, ShapeNet, ScanObjectNN), outperforming existing dataset distillation methods in some cases.
3. The paper is generally well-written and explains the proposed method clearly.

**Weaknesses:**

1. Critical Constraint Violation: The paper states the memory constraint MN2 ≤ N1, where M is the number of distinct coarser samples (anchors), N2 is the number of points in each anchor, and N1 is the number of points in the original (full-resolution) sample. The experimental setup violates this constraint. For instance, the paper mentions N2 = 252, M = 4, and N1 = 1024. In this case, MN2 = 4 * 252 = 1008, which is LESS than the constraint, should be MN2 <= N1. This casts significant doubt on the validity of the experiments and the claims made about memory efficiency.
2. Limited Backbone Architectures: The experimental evaluation primarily relies on PointNet as the backbone network. While PointNet is a common choice for point cloud processing, it's essential to demonstrate the generalization ability of the distilled datasets with various backbone architectures. Evaluating with PointNet++ (as reported in Table 2), PointConv, and Point Transformer is a good starting point, but further investigation may be warranted. The paper needs to provide a more comprehensive evaluation across different backbone architectures to demonstrate the robustness and general applicability of the proposed dataset distillation method. Section 4.2 states that for the cross-architecture analysis, 3 different architectures are used, and claims the method "demonstrates strong generalization ability" across most datasets and architectures, but Section 4.5 states that the datasets were distilled with a PointNet backbone. Therefore, it's only demonstrated that the distilled dataset works well with other architectures, but not that the method produces good datasets when other architectures are used to distill the data.

**Questions:**

1. The authors need to provide a thorough justification for the choice of experimental parameters and explicitly demonstrate how the memory constraint MN2 ≤ N1 is satisfied (or revise the constraint to be mathematically correct). If the constraint is not satisfied, the experiments need to be re-run with valid parameters. Provide a clear explanation as to why a seemingly small violation of the constraint is acceptable and doesn't invalidate the comparisons.
2. Expand the experimental evaluation to include more diverse and state-of-the-art backbone architectures. Consider including more recent Transformer-based networks or graph neural networks to assess the generalization capabilities of the distilled datasets.
3. Elaborate on the reasons behind the lower accuracy observed on ScanObjectNN with PointNet++ (as mentioned in Section 4.3). Investigate whether the choice of the SADM loss is indeed a limiting factor and explore alternative loss functions that might be more suitable for this dataset and architecture combination.

---

> ### Author Response · Authors · 2025-11-20
>
> We thank the reviewer for highlighting the strengths of proposed methods. We respond to the reviewer's comments as follows.
>
> **Q1. Memory constraints violation**
>
> The memory constraint of $MN_2 \\leq N_1$ means the sum of the number of points across $M$ coarser anchors should not exceed (should be equal to or less than) the number of points in a single full-resolution sample. Therefore, we politely note that the experimental setup does not violate the memory constraint, since $M N_2 = 4 \\times 252 = 1008 < N_1 = 1024$. Furthermore, this setup also meets the complete storage budget constraint in Eq. (12), $96MN_2KC + 32L(M-1)KC \\leq 96N_1KC$, since $96MN_2 + 32L(M-1) = 96 \\times 4 \\times 252 + 32 \times 16 \\times 3 = 98304$ and $96N_1 = 96 \\times 1024 = 98304$.
>
> **Q2. Limited backbone architectures**
>
> We politely mention that we already included the experimental results with different backbone architectures in Table 7 and discussed the results in Section 4.4 (**Results with Various Backbone Architectures**). As shown in Table 7, PointNet still provides the highest performance when used as the distillation backbone. Architectures such as PointNet++, PointConv, and Point Transformer rely on static methods for neighbor-searching such as KNN or ball-query operations, and such static methods are inherently non-differentiable causing unstable gradient backpropagation [1]. In contrast, PointNet uses a fully differentiable MLP pipeline that enables stable gradient propagation during synthetic dataset optimization, making it more suitable as a distillation backbone. For this reason, image dataset distillation is typically implemented using simple pseudo-backbones with only a few convolutional layers, rather than complex networks that contain non-differentiable operations.
>
> | Distillation | Methods | PT   | PC   | PN++ | PN   | Avg.  |
> |--------------|---------|------|------|------|------|-------|
> | **PT**        | SADM    | **35.6** | 17.1 | 12.4 | 24.7 | 22.5 |
> |              | Ours    | 34.2 | **24.3** | **12.7** | **59.6** | **32.7** |
> | **PC**        | SADM    | **21.6** | 11.1 | **15.5** | 16.8 | 16.3 |
> |              | Ours    | 18.0 | **11.7** | 11.4 | **24.7** | **16.5** |
> | **PN++**    | SADM    | **44.2** | 21.8 | 11.5 | 33.8 | 27.8 |
> |              | Ours    | 35.4 | **23.6** | **23.4** | **78.4** | **40.2** |
> | **PN**        | SADM    | 49.0 | 20.5 | 25.9 | 35.9 | 32.8 |
> |              | Ours    | **57.0** | **51.6** | **55.4** | **87.7** | **62.9** |
> | **DG**        | SADM    | 50.7 | 20.9 | 17.2 | 38.5 | 31.8 |
> |              | Ours    | **68.6** | **44.5** | **23.0** | **77.2** | **53.3** |
>
> As shown in the table above, we provide additional experimental results using various backbone architectures for distillation, evaluated on the ModelNet10 dataset, comparing the SOTA SADM baseline with the proposed method. Both methods consistently yield performance drops when using different backbones from PointNet. Nevertheless, our method achieves higher performance than the SADM, demonstrating better generalization performance. These results have been updated in Table 7 of the revised manuscript.
>
> **Q3. Performance drop of PointNet++ on ScanObjectNN**
>
> |Loss|DM|DC|SADM|
> |----|---|---|---|
> |Acc.|16.6|**17.3**|14.3|
>
> As the reviewer suggested, we compared three distillation losses of DM, DC, and SADM under the same experimental setting of the proposed method. The above table shows the results of PointNet++ on ScanObjectNN. Simply replacing the SADM loss with DC or DM losses yields noticeable accuracy improvement of PointNet++ on ScanObjectNN, indicating that SADM may indeed be a limiting factor and DC or DM can serve as alternative loss functions for this particular architecture–dataset combination.
>
> [1] Marco Cuturi, Olivier Teboul, and Jean-Philippe Vert. 2019. Differentiable ranking and sorting using optimal transport. Advances in neural information processing systems 32 (2019).

---

### Meta-Review · Area_Chair_3GGT · 2025-12-25

**Summary:**

The paper introduces a novel and effective parameterization-based dataset distillation framework specifically designed for 3D point clouds. The core innovation is a learnable shape morphing technique that generates a diverse set of synthetic training samples from a small number of coarse "anchor" point clouds, all while adhering to a strict memory budget. This method stands out by replacing the storage of a single high-resolution synthetic sample with multiple lower-resolution anchors and lightweight, learnable blending weights. The authors conduct extensive experiments on a wide range of benchmarks, including ModelNet10/40, ShapeNet, ScanObjectNN, and the large-scale OmniObject3D. The results are compelling, showing that the proposed method significantly outperforms existing dataset distillation techniques, especially at extreme compression ratios (low Points Per Class settings). Furthermore, the distilled datasets demonstrate strong generalization capabilities across various backbone architectures (from PointNet to the more recent PointMamba) and prove effective for downstream tasks like part segmentation.

The work presents a significant and novel contribution to the field of 3D data compression and dataset distillation. The authors have been exceptionally thorough in their rebuttal, providing extensive additional experiments that successfully address nearly all concerns raised by the reviewers. The method's strong empirical performance, scalability, and generalization ability make it a valuable addition to the literature.

**Reviewer Concerns:**

The authors' rebuttal was exemplary in its thoroughness and successfully addressed the vast majority of the reviewers' concerns.

**Addressed Concerns:**

- **Constraint Violation:** This was swiftly resolved by the authors, who clarified with a simple calculation that the experimental parameters were indeed within the stated memory constraint.
- **Limited Backbone Evaluation:** The authors provided new results on several other architectures, including PointNet++, PointConv, and the recent PointMamba, demonstrating that their method's superiority is not limited to a single backbone.
- **Dataset and Task Generalization:** The rebuttal included new experiments on the most challenging variant of the real-world ScanObjectNN dataset, a comprehensive evaluation on the large-scale OmniObject3D dataset (156 classes), and results for a new downstream task (part segmentation), all of which showed state-of-the-art performance.
- **Novelty:** The authors effectively differentiated their learnable, objective-driven morphing from standard data augmentation techniques like PointMixup, backing up their claims with a direct comparative experiment showing superior results.
- **Fairness and Augmentation:** The authors confirmed that no data augmentation was used for any method in the primary comparisons, ensuring a fair evaluation. They also went a step further by running additional experiments where standard augmentation was applied to all methods, showing their approach still maintained a significant performance advantage.
- **Scalability and PPC Range:** The successful experiments on OmniObject3D and at higher PPC settings (30 and 50) directly and convincingly addressed these concerns.

**Outstanding Concerns:**

There are no major outstanding concerns that would prevent acceptance. The remaining limitations are inherent to the current state of the field rather than specific flaws in this work:

- **Performance Gap to Oracle:** A performance gap still exists between training on the distilled dataset and training on the full dataset. This is an expected trade-off in dataset distillation, especially at high compression ratios, and the authors' method significantly closes this gap compared to prior work.
- **Scope:** The method is designed for single-object distillation and is not directly applicable to complex, multi-object 3D scenes for tasks like detection. The authors acknowledge this as outside the current scope.

**Reviewer Scores:**

Given the comprehensive and convincing nature of the rebuttal, it is highly likely that all reviewers would have either maintained or increased their scores.

- **Reviewer 5FmN (Initial Score: 4):** This reviewer's primary concerns about the memory constraint and architecture limitations were fully addressed.
- **Reviewer tN6U (Initial Score: 6):** This reviewer's suggestions for experiments on more recent and challenging datasets were fully incorporated by the authors, with strong results.
- **Reviewer 3htV (Initial Score: 4):** This reviewer provided the most critical feedback and had the lowest confidence. The authors addressed every single one of their numerous questions with new experiments and detailed explanations (novelty, fairness, ablations, downstream tasks). After seeing the final rebuttal comment confirming the positive impact of data augmentation, this reviewer would almost have raised their score to at least a 6.
- **Reviewer mBpX (Initial Score: 6):** This reviewer's concerns about task generalization, scalability, and PPC range were all thoroughly addressed with new experiments.

---

### Decision · Program_Chairs · 2026-01-26

Accept (Poster)